# Predicted glycosyltransferases promote development and prevent spurious cell clumping in the choanoflagellate *S. rosetta*

Laura A Wetzel[1,2], Tera C Levin[1,2], Ryan E Hulett[1,2], Daniel Chan[1,2], Grant A King[1,2], Reef Aldayafleh[1,2], David S Booth[1,2], Monika Abedin Sigg[1,2], Nicole King[1,2]*

[1]Department of Molecular and Cell Biology, University of California, Berkeley, Berkeley, United States; [2]Howard Hughes Medical Institute, University of California, Berkeley, Berkeley, United States

**Abstract** In a previous study we established forward genetics in the choanoflagellate *Salpingoeca rosetta* and found that a C-type lectin gene is required for rosette development (Levin *et al.*, 2014). Here we report on critical improvements to genetic screens in *S. rosetta* while also investigating the genetic basis for rosette defect mutants in which single cells fail to develop into orderly rosettes and instead aggregate promiscuously into amorphous clumps of cells. Two of the mutants, Jumble and Couscous, mapped to lesions in genes encoding two different predicted glycosyltransferases and displayed aberrant glycosylation patterns in the basal extracellular matrix (ECM). In animals, glycosyltransferases sculpt the polysaccharide-rich ECM, regulate integrin and cadherin activity, and, when disrupted, contribute to tumorigenesis. The finding that predicted glycosyltransferases promote proper rosette development and prevent cell aggregation in *S. rosetta* suggests a pre-metazoan role for glycosyltransferases in regulating development and preventing abnormal tumor-like multicellularity.
DOI: https://doi.org/10.7554/eLife.41482.001

*For correspondence:
nking@berkeley.edu

## Introduction

The transition to multicellularity was essential for the evolution of animals from their single celled ancestors (*Szathmáry and Smith, 1995*). However, despite the centrality of multicellularity to the origin of animals, little is known about the genetic and developmental mechanisms that precipitated the evolution of multicellularity on the animal stem lineage. All modern animals develop clonally through serial cell division, suggesting that the same was true for their last common ancestor. While the closest living relatives of animals, choanoflagellates, develop clonally into multicellular rosettes, more distant relatives such as *Capsaspora owczarzaki* (*Sebé-Pedrós et al., 2013*) and *Dictyostelium discoideum* (*Bonner, 1967*; *Brefeld, 1869*) become multicellular through cell aggregation (which is vulnerable to cheating) (*Santorelli et al., 2008*; *Strassmann et al., 2000*). This raises a general question of how stem animals might have suppressed cell aggregation in favor of clonal multicellular development.

Although the first animals evolved over 600 million years ago, studying choanoflagellates allows the reconstruction of important aspects of animal origins (*Brunet and King, 2017*; *King et al., 2008*; *Ruiz-Trillo et al., 2008*; *Shalchian-Tabrizi et al., 2008*; *Sebé-Pedrós et al., 2017*). *Salpingoeca rosetta* is an emerging model choanoflagellate that was isolated from nature as a spherical colony of cells called a rosette. Under standard laboratory conditions, *S. rosetta* proliferates as

solitary cells or as linear chain colonies that easily break apart into solitary cells (*Dayel et al., 2011*). When exposed to rosette inducing factors (RIFs) produced by the co-isolated prey bacterium *Algoriphagus machipongonensis, S. rosetta* instead develops into highly organized and structurally stable rosettes through a process of serial cell division (*Alegado et al., 2012*; *Dayel et al., 2011*; *Fairclough et al., 2010*; *Woznica et al., 2016*). Recent advances, including a sequenced genome (*Fairclough et al., 2010*), the discovery of a sexual phase to the *S. rosetta* life cycle that enables controlled mating (*Levin et al., 2014*; *Levin and King, 2013*; *Woznica et al., 2017*), and techniques that allow for transfection and expression of transgenes (*Booth et al., 2018*) have enabled increasingly detailed studies of molecular mechanisms underlying rosette development in *S. rosetta*.

In the first genetic screen to identify genes required for rosette formation in *S. rosetta*, multiple rosette defect mutants were recovered that displayed a range of phenotypes (*Levin et al., 2014*). The first mutant to be characterized in detail was named Rosetteless; while Rosetteless cells did not develop into rosettes in the presence of RIFs, they were otherwise indistinguishable from wild type cells (*Levin et al., 2014*). The mutation underlying the Rosetteless phenotype was mapped to a C-type lectin, encoded by the gene *rosetteless,* the first gene shown to be required for rosette formation (*Levin et al., 2014*). In animals, C-type lectins function in signaling and adhesion to promote development and innate immunity (*Cambi et al., 2005*; *Geijtenbeek and Gringhuis, 2009*; *Ruoslahti, 1996*; *Svajger et al., 2010*; *Zelensky and Gready, 2005*). Although the molecular mechanisms by which *rosetteless* regulates rosette development remain unknown, the localization of Rosetteless protein to the rosette interior suggests that it functions as part of the extracellular matrix (ECM) (*Levin et al., 2014*).

Here we report on the largest class of mutants from the original rosette defect screen (*Levin et al., 2014*), all of which fail to develop into organized rosettes and instead form large, amorphous clumps of cells in both the absence and presence of RIFs. By mapping the mutations underlying the clumpy, rosette defect phenotypes of two mutants in this class, we identified two predicted glycosyltransferase genes that are each essential for proper rosette development. The causative mutations led to similar perturbations in the glycosylation pattern of the basal ECM. The essentiality of the predicted glycosyltransferases for rosette development combined with prior findings of the requirement of a C-type lectin highlight the importance of the ECM for regulating multicellular rosette development and preventing spurious cell adhesion in a close relative of animals.

## Results

### Rosette defect mutants form amorphous clumps of cells through promiscuous cell adhesion

The original rosette defect screen performed by *Levin et al. (2014)* yielded nine mutants that were sorted into seven provisional phenotypic classes. For this study, we screened 21,925 additional clones and identified an additional seven mutants that failed to form proper rosettes in the presence of *Algoriphagus* RIFs. (For this study, we used *Algoriphagus* outer membrane vesicles as a source of RIFs, as described in *Woznica et al., 2016*). Comparing the phenotypes of the 16 total rosette defect mutants in the presence and absence of RIFs allowed us to classify four broad phenotypic classes: (1) Class A mutants that have wild type morphologies in the absence of RIFs and entirely lack rosettes in the presence of RIFs, (2) Class B mutants that have wild type morphologies in the absence of RIFs and develop reduced levels of rosettes with aberrant structures in the presence of RIFs, (3) Class C mutants that produce large clumps of cells in both the presence and absence of RIFs while forming little to no rosettes in the presence of RIFs, and (4) a Class D mutant that exist primarily as solitary cells, with no linear chains of cells detected in the absence of RIFs and no rosettes detected in the presence of RIFs (*Supplementary file - Table S1*).

Of the 16 rosette defect mutants isolated, seven mutants fell into Class C. For this study, we focused on four Class C mutants — Seafoam, Soapsuds, Jumble, and Couscous (previously named Branched in *Levin et al., 2014*) — that form amorphous, tightly packed clumps of cells, both in the presence and absence of RIFs, but never develop into rosettes (*Table 1*; *Figure 1A,B*). We found that the clumps contain a few to hundreds of mutant cells that pack together haphazardly, unlike

**Table 1.** Phenotypes of wild type and Class C mutants

| Strain | % cells in rosettes | Cell interactions | Successful outcross? |
|---|---|---|---|
| wild type | 87.7 | Non-clumping | Yes |
| Seafoam | 0 | Clumping | No |
| Soapsuds | 0 | Clumping | No |
| Couscous | 0 | Clumping | Yes |
| Jumble | 0 | Clumping | Yes |

DOI: https://doi.org/10.7554/eLife.41482.007

wild type rosettes in which all cells are oriented with their basal poles toward the rosette center and their apical flagella extending out from the rosette surface (*Alegado et al., 2012*; *Levin et al., 2014*; *Woznica et al., 2016*). Moreover, in contrast with the structural stability and shear resistance of wild type rosettes (*Figure 1A*) (*Levin et al., 2014*), the cell clumps formed by Class C mutants were sensitive to shear and separated into solitary cells upon pipetting or vortexing the culture (*Figure 1A*).

Following exposure to shear, we observed that mutant cells re-aggregated into new clumps within minutes, while wild type cells never formed clumps (*Figure 1C,D*; rare cell doublets were likely due to recent cell divisions). Within 30 min after disruption by shear force, cell clumps as large as 75, 55, 32, and 23 cells formed in Couscous, Soapsuds, Seafoam, and Jumble mutant cultures, respectively. Moreover, blocking cell division with the cell cycle inhibitor aphidicolin did not prevent clump formation (*Figure 1—figure supplement 1*). Both the speed of clump reformation (less than the ~6 hr required for a single cell division (*Levin et al., 2014*; *Figure 1—figure supplement 2*) and the observation of cell clumping in the absence of cell division (*Figure 1—figure supplement 1*) demonstrate that cell aggregation alone is sufficient to drive clump formation. Indeed, each of the mutants tested also displayed a mild defect in cell proliferation (*Figure 1—figure supplement 2*).

Therefore, the cell clumps are not aberrant rosettes, which never form through aggregation and instead require at least 15–24 hr to develop clonally through serial rounds of cell division (*Dayel et al., 2011*; *Fairclough et al., 2010*). Nonetheless, we tested whether Jumble and Couscous clump formation might be influenced by the presence or absence of RIFs. Clumps formed in both the presence and absence of RIFs were comparable in size (K-S test; *Figure 1—figure supplement 3*). Cell aggregation was not strain-specific, as unlabeled Jumble and Couscous mutant cells adhered to wild type cells identified by their expression of cytoplasmic mWasabi (*Figure 1—figure supplement 4*).

The fact that the seven clumping/aggregating Class C mutants isolated in this screen were also defective in rosette development suggests a direct link between promiscuous cell adhesion and failed rosette development.

## Improving genetic mapping in *S. rosetta* through bulk segregant analysis

We next set out to identify the causative mutation(s) underlying the clumping and rosette defect phenotypes in each of these mutants. In the *Levin et al. (2014)* study, the Rosetteless mutant was crossed to a phenotypically wild type Mapping Strain (previously called Isolate B in *Levin et al., 2014*) and relied on genotyping of haploid F1s at 60 PCR-verified genetic markers that differed between the Rosetteless mutant and the Mapping Strain (*Levin et al., 2014*). The 60 markers were distributed unevenly across the 55 Mb genome and proved to be insufficient for mapping the Class C mutants for this study. Compounding the problem, the low level of sequence polymorphism among *S. rosetta* laboratory strains and abundance of repetitive sequences in the draft genome assembly (*Fairclough et al., 2013*; *Levin et al., 2014*) made it difficult to identify and validate additional genetic markers, while genotyping at individual markers proved labor intensive and costly.

To overcome these barriers, we modified bulk segregation methods developed in other systems (*Doitsidou et al., 2010*; *Leshchiner et al., 2012*; *Lister et al., 2009*; *Pomraning et al., 2011*; *Schneeberger et al., 2009*; *Voz et al., 2012*; *Wenger et al., 2010*) for use in *S. rosetta*. Our strategy involved: (1) crossing mutants to the Mapping Strain (which contains previously identified

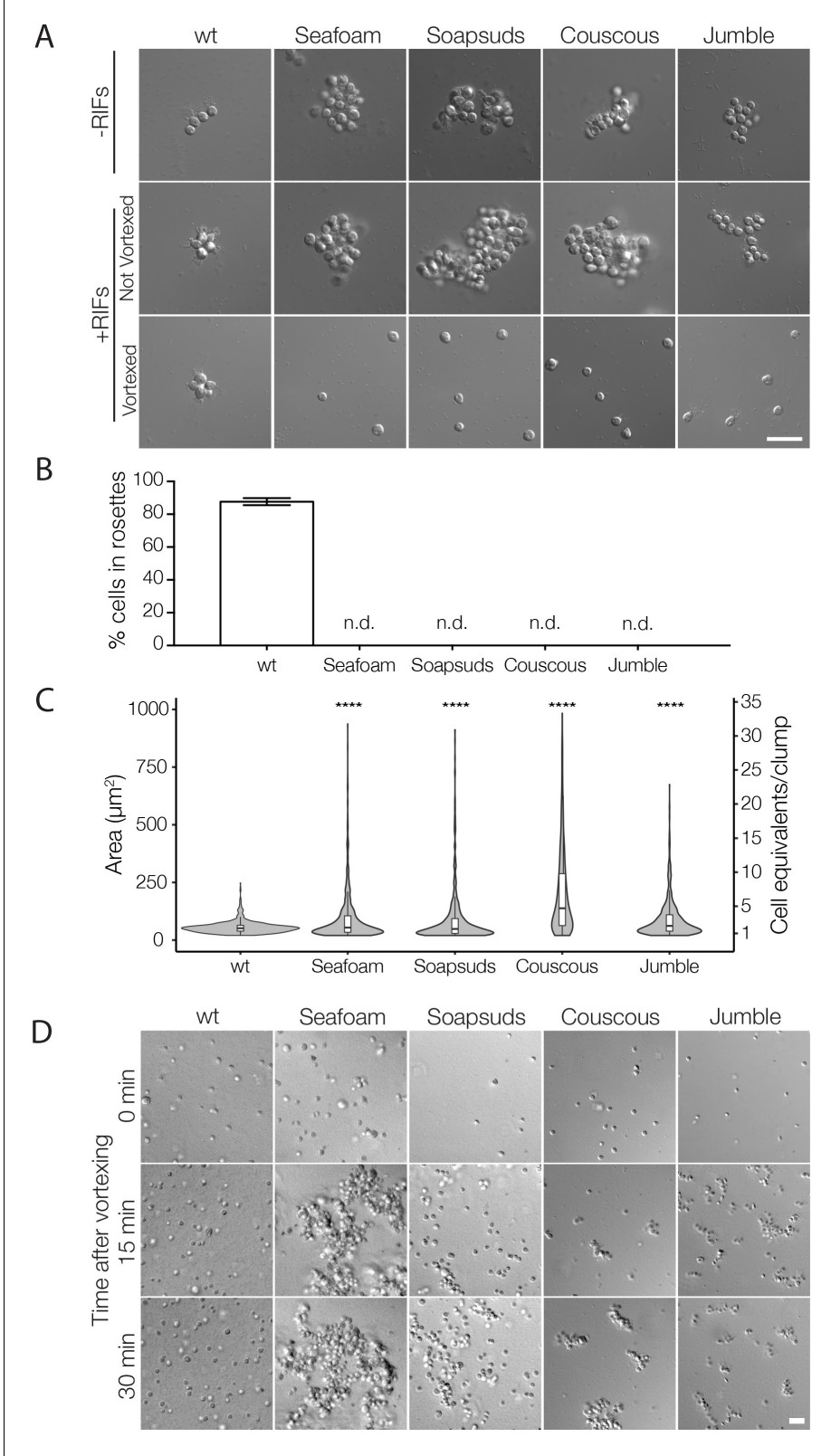

**Figure 1.** Mutant cells aggregate and fail to form rosettes. (**A**) Wild type cells are unicellular or form linear chains in the absence of rosette inducing factors (RIFs) and develop into organized spherical rosettes when cultured with RIFs. Rosettes are resistant to shear force and survive vortexing. Four class C mutants — Seafoam, Soapsuds, Couscous, and Jumble — form disordered clumps of cells in the presence and absence of RIFs. The clumps are not resistant to vortexing and fall apart into single cells. (**B**) Class C mutants do not form any detectable rosettes. Rosette development was measured

*Figure 1 continued on next page*

*Figure 1 continued*

as the % of cells in rosettes after 48 hr in the presence of RIFs and is shown as mean ± SEM. n.d. = no detected rosettes. (**C**) Class C mutants quickly aggregated into large clumps after disruption by vortexing. After vortexing, wild type and mutant cells were incubated for 30 min in the absence of RIFs and clump sizes were quantified by automated image analysis. Data are presented as violin boxplots, showing the median cell number (horizontal line), interquartile range (white box), and range excluding outliers (vertical line). All mutants had significantly larger masses of cells (K-S test, ****p < 0.0001) than found in cultures of wild type cells. (**D**) Clumping occurred within minutes after vortexing in the Class C mutants without RIFs, revealing that the clumps form by aggregation and not through cell division. DIC images obtained at 0, 15, and 30 min post-vortexing. Scale bars = 20 μm.

DOI: https://doi.org/10.7554/eLife.41482.002

The following figure supplements are available for figure 1:

**Figure supplement 1.** Cell division is not required for clump formation in mutants.

DOI: https://doi.org/10.7554/eLife.41482.003

**Figure supplement 2.** Class C mutant growth curves.

DOI: https://doi.org/10.7554/eLife.41482.004

**Figure supplement 3.** Jumble and Couscous clumps formed in the absence or presence of RIFs are comparable in size.

DOI: https://doi.org/10.7554/eLife.41482.005

**Figure supplement 4.** Jumble and Couscous cells adhere to wild type cells.

DOI: https://doi.org/10.7554/eLife.41482.006

sequence variants); (2) isolating heterozygous diploids identified through genotyping at a microsatellite on supercontig 1; (3) inducing meiosis; (4) growing clonal cultures of haploid F1 offspring; (5) phenotyping the F1 offspring; (6) pooling F1 offspring based on their clumping phenotype; and (7) deeply sequencing pooled genomic DNA from the F1 mutants to find mutations that segregated with the clumping phenotype (*Figure 2—figure supplement 1*).

To test whether a bulk segregant approach would work in *S. rosetta*, we first analyzed a cross between the previously mapped Rosetteless mutant and the Mapping Strain (*Levin et al., 2014*). We isolated 38 F1s with the rosette defect phenotype from a Mapping Strain × Rosetteless cross (*Levin et al., 2014*), grew clonal cultures from each, pooled the resulting cultures, extracted their genomic DNA, and sequenced the pooled mutant genomes to an average coverage of 187X. Against a background of sequence variants that did not segregate with the Rosetteless phenotype, five unlinked single nucleotide variants (SNVs) and insertions/deletions (INDELs) were found to segregate with the phenotype (*Supplementary file - Table S2*). Four of these detected sequence variants likely had spurious correlations with the phenotype resulting from relatively low sequencing coverage at those variants (>0.25X coverage of the entire genome) (*Supplementary file - Table S2*). In contrast, the remaining SNV was detected in a well-assembled portion of the genome at a sequencing depth approaching the average coverage of the entire genome. The segregating SNV, at position 427,804 on supercontig 8, was identical to the causative mutation identified in *Levin et al. (2014)* (*Supplementary file - Table S2*). Thus, a method based on pooling F1 haploid mutants, identifying sequence variants that segregated with the phenotype, and masking those SNVs/INDELs that were detected with >0.25X coverage of the total genome was effective for correctly pinpointing the causal mutation for Rosetteless (*Figure 2—figure supplement 1*). Therefore, we used this validated bulk segregant method to map the clumping mutants.

Mapping crosses were carried out for the four clumping/rosette defect mutants characterized in this study (Seafoam, Soapsuds, Jumble, and Couscous) and all four crosses yielded heterozygous diploids, demonstrating that they were competent to mate. As observed in prior studies of *S. rosetta* mating (*Levin et al., 2014*; *Woznica et al., 2017*), the diploid cells each secreted a flask-shaped attachment structure called a theca and were obligately unicellular. Therefore, the heterozygous diploids were not informative about whether the mutations were dominant or recessive as the phenotypes could only be detected in haploid cells. For Seafoam and Soapsuds, we isolated heterozygous diploids, but never recovered F1 offspring with the mutant phenotype (*Table 1*). The inability to recover haploids with either clumping or rosette defect phenotypes from the Seafoam × Mapping Strain and Soapsuds × Mapping Strain crosses might be explained by any of the following: (1) the clumping/rosette defect phenotypes are polygenic, (2) meiosis defects are associated with the causative mutations, and/or (3) mutant fitness defects allowed wild type progeny to outcompete the mutant progeny. In contrast, heterozygous diploids from crosses of Jumble and Couscous to the

Mapping Strain produced F1 haploid progeny with both wild type and mutant phenotypes and thus allowed for the successful mapping of the causative genetic lesions, as detailed below.

## Jumble maps to a putative glycosyltransferase

Following the bulk segregant approach, we identified five sequence variants in Jumble that segregated with both the clumping and rosette defects. Only one of these – at position 1,919,681 on supercontig 1 – had sequencing coverage of at least 0.25X of the average sequence coverage of the rest of the genome (*Figure 2A*; *Supplementary file - Table S3*). In a backcross of mutant F1 progeny to the Mapping Strain, we confirmed the tight linkage of the SNV to the rosette defect phenotype (*Figure 2B*). Moreover, all F2 progeny that displayed a rosette defect also had a clumping phenotype. Given the tight linkage of both traits with the SNV and the absence of any detectable neighboring sequence variants, we infer that the single point mutation at genome position 1:1,919,681 causes both the clumping and rosette defect phenotypes in Jumble mutants.

The mutation causes a T to C transition in a gene hereafter called *jumble* (GenBank accession EGD72416/NCBI accession XM_004998928; *Figure 2A*). The *jumble* gene contains a single exon and is predicted to encode a 467 amino acid protein containing a single transmembrane domain. Following the convention established in *Levin et al. (2014)*, the mutant allele, which is predicted to confer a leucine to proline substitution at amino acid position 305, is called *jumble^lw1*.

We used recently developed methods for transgene expression in *S. rosetta* (*Booth et al., 2018*) to test whether expression of a *jumble* with an N- or C-terminal *monomeric teal fluorescent protein* (*mTFP*) gene fusion under the *S. rosetta elongation factor L* (*efl*) promoter could complement the mutation and rescue rosette development in the Jumble mutant (*Figure 2C,D*). We were able to enrich for rare transformed cells by using a plasmid in which the puromycin resistance gene (*pac*) was expressed under the same promoter as the *jumble* fusion gene, with the two coding sequences separated by a sequence encoding a self-cleaving peptide (*Kim et al., 2011*). Transfection of Jumble mutant cells with wild type *jumble-mTFP* followed by puromycin selection and the addition of RIFs yielded cultures in which 9.33 ± 5.07% of cells were in rosettes (*Figure 2C*). Similarly, transfection of Jumble with *mTFP-jumble* followed by puromycin selection and rosette induction resulted in cultures with 7.00 ± 4.91% of cells in rosettes (*Figure 2C*). Importantly, we did not detect any rosettes when we transfected Jumble cells with *mTFP* alone, *jumble^lw1-mTFP*, or *mTFP-jumble^lw1*. Complementation of the Jumble mutant by the wild type *jumble* allele, albeit in a subset of the population, provided further confirmation that the *jumble^lw1* mutation causes the cell clumping and rosette defect phenotypes. The fact that the transfection experiment did not allow all cells to develop into rosettes may be due to any number of reasons, including incomplete selection against untransformed cells, differences in transgene expression levels in different transformed cells, and the possibility that the mTFP tag reduces or otherwise changes the activity of the Jumble protein.

We next sought to determine the function and phylogenetic distribution of the *jumble* gene. BLAST searches uncovered unannotated *jumble* homologs in nine other choanoflagellates (*Figure 2—figure supplement 2A*) and in fungi (*Figure 2—figure supplement 3*), but none in animals. The choanoflagellate homologs of *jumble* were detected in the transcriptomes of species representing each of the three major choanoflagellate clades (*Richter et al., 2018*), suggesting that *jumble* evolved before the origin and diversification of choanoflagellates. Although Interpro (*Finn et al., 2017*) and Pfam (*Finn et al., 2016*) did not reveal any known protein domains in Jumble, the NCBI Conserved Domain Search (*Marchler-Bauer et al., 2017*) predicted a glycosyltransferase domain with low confidence (E-value $3.87^{-03}$). Moreover, two different algorithms that use predicted secondary and tertiary structures to identify potential homologs, HHphred (*Zimmermann et al., 2018*) and Phyre2 (*Kelley et al., 2015*), predict that Jumble is related to well-annotated glycosyltransferases (HHphred: E-value $7.5^{-19}$ to polypeptide N-acetylgalactosaminyltransferase 4; Phyre2: Confidence 94.5% to human polypeptide n-acetylgalactosaminyltransferase 2) (*Figure 2—figure supplement 2B*). The Leu305Pro substitution in Jumble^lw1 disrupts a predicted alpha helix, which we hypothesize would prevent proper folding of the Jumble protein (*Figure 2A*).

Glycosyltransferases play essential roles in animal development (*Sawaguchi et al., 2017*; *Zhang et al., 2008*) and cell adhesion (*Müller et al., 1979*; *Stratford, 1992*). Their biochemical functions include transferring an activated nucleotide sugar, also called a glycosyl donor, to lipid, protein, or carbohydrate acceptors (*Lairson et al., 2008*). Target acceptors in animals include key signaling and adhesion proteins such as integrins and cadherins, whose activities are regulated by N-

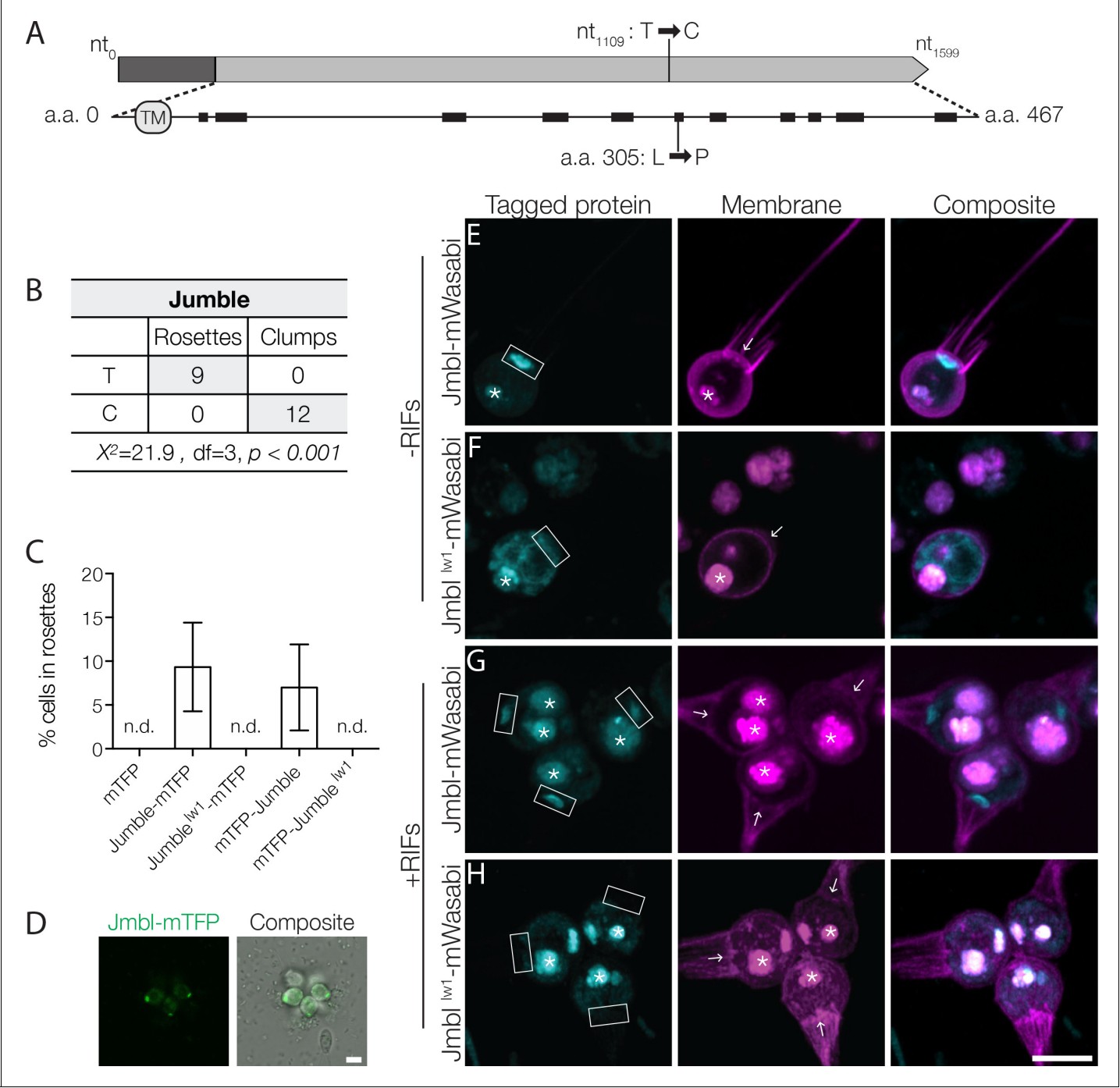

**Figure 2.** Jumble maps to a predicted glycosyltransferase that localizes to the Golgi apparatus. (**A**) Jumble has a predicted transmembrane domain (marked TM) and secondary structure (alpha helices marked by black rectangles). Structural homology algorithms predict that Jumble is structurally related to well-characterized glycosyltransferases (*Figure 2—figure supplement 2B*). The mutant gene has a T to C mutation at nucleotide 1109 that causes an amino acid substitution of proline to leucine at amino acid position 305. (**B**) A backcross of a mutant F1 progeny to the Mapping Strain yielded nine rosette-forming F2 isolates with the wild type T allele and twelve clumpy F2 isolates with the *jumble^{lw1}* C allele. The inheritance significantly deviated from expected Mendelian inheritance of unlinked traits and confirmed the tight linkage between the *jumble^{lw1}* allele and the clumpy, rosetteless phenotype. $X^2$ = Chi squared value, d.f. = degrees of freedom. (**C,D**) Transgenic expression of *jumble-mTFP* and *mTFP-jumble* rescued rosette development in the Jumble mutant, but *jumble^{lw1}-mTFP*, *mTFP-jumble^{lw1}*, or *mTFP* did not. RIFs were added immediately after transfection and 40 µg/ml puromycin was added 24 hr post-transfection to select for transformants. (**C**) Rosette development was measured as the % of cells in rosettes 72 hr post-transfection and shown as mean ± SD. n.d. = no detected rosettes. (n = 200 cells counted from each of 3 technical replicates; two biological replicates). (**D**) Rosettes transgenically complemented with *jumble-mTFP* in the Jumble mutant appeared phenotypically wild

*Figure 2 continued on next page*

*Figure 2 continued*

type and most cells in rosettes had detectable fluorescent expression at the apical base of the cell. Representative rosette shown. (**E–H**) To examine localization, Jumble-mWasabi or Jumble[lw1]-mWasabi (cyan) under the *efl* promoter were co-expressed with membrane marker-mCherry (magenta) in wild type *S. rosetta*. Jumble-mWasabi localizes to the apical pole of cells grown (**E**) without RIFs or (**G**) with RIFs, consistent with the localization of the Golgi apparatus. When expressed in otherwise wild type cells grown (**F**) without RIFs or (**H**) with RIFs, the mutant Jumble[lw1]-mWasabi incorrectly localizes to the ER and food vacuole. Boxes indicate the inferred location of the Golgi apparatus at the apical pole of the cell. The food vacuole (asterisk) was often visualized due to autofluoresence from ingested bacteria or through accumulation of the fluorescent markers in the food vacuole, perhaps through autophagy. For reference, arrows indicate the base of the flagellum although the flagellum may not be visible in the plane of focus shown. Scale bars = 5 µm.

DOI: https://doi.org/10.7554/eLife.41482.008

The following figure supplements are available for figure 2:

**Figure supplement 1.** Mapping cross scheme.

DOI: https://doi.org/10.7554/eLife.41482.009

**Figure supplement 2.** Alignment of Jumble homologs and predicted structure.

DOI: https://doi.org/10.7554/eLife.41482.010

**Figure supplement 3.** Alignment of Jumble to fungal homologs.

DOI: https://doi.org/10.7554/eLife.41482.011

**Figure supplement 4.** Ultrastructure of *S.rosetta* and ER co-localization of Jumble[lw1].

DOI: https://doi.org/10.7554/eLife.41482.012

and O-linked polysaccharide modifications, also referred to as N- and O-linked glycans (*Larsen et al., 2017*; *Zhao et al., 2008*). Notably, many well-characterized glycosyltransferases act in the Golgi apparatus, where they glycosylate molecules that are trafficked through the secretory system (*El-Battari, 2006*; *Tu and Banfield, 2010*). To investigate the localization of Jumble, we transfected wild type cells with a *jumble-mWasabi* gene fusion transcribed under the control of the *S. rosetta efl* promoter. Jumble-mWasabi protein localized to the apical pole of the cell body near the base of the flagellum. Based on comparisons with transmission electron micrographs of *S. rosetta* and other choanoflagellates, Jumble-mWasabi localization corresponds to the location of the Golgi apparatus, for which there is not yet a fluorescent marker in *S. rosetta* (*Figure 2E,G*; *Figure 2—figure supplement 4A*) (*Leadbeater, 2015*). In contrast, Jumble[lw1]-mWasabi, was distributed in a tubular pattern throughout the cell and co-localized with an endoplasmic reticulum (ER) marker (*Figure 2F,H*; *Figure 2—figure supplement 4B*) (*Booth et al., 2018*). The ER localization of Jumble[lw1] is consistent with the hypothesis that the missense mutation disrupts proper protein folding as often misfolded proteins are retained in the ER and targeted for degradation (*Kopito, 1997*). The failure of the Jumble[lw1] protein to localize properly at the Golgi apparatus strongly suggests a loss of function.

## Couscous maps to a lesion in a predicted mannosyltransferase

We followed a similar strategy to map the genetic lesion(s) underlying the Couscous mutant phenotype. Using the bulk segregant approach on F1 mutant offspring from a Couscous × Mapping Strain cross, we identified eight sequence variants that segregated with the clumping and rosette defect phenotypes, of which only one – a single nucleotide deletion at position 462,534 on supercontig 22 – had sequencing coverage at least 0.25X of the average sequence coverage of the rest of the genome (*Figure 3A*; *Supplementary file - Table S4*). The tight linkage of the deletion to both the clumping and rosette defect phenotypes was further confirmed by genotyping the sequence variant in F2 mutants resulting from backcrosses of F1 mutants to the Mapping Strain (*Figure 3B*). Given the tight linkage, we infer that the deletion at position 462,534 on supercontig 22 causes both clumping and the disruption of rosette development in Couscous mutant cells.

The single nucleotide deletion at position 462,534 on supercontig 22 lies in a four-exon gene, hereafter called *couscous* (GenBank accession EGD77026/NCBI accession XM_004990809). The mutation causes a predicted frameshift leading to an early stop codon in the mutant protein, Couscous[lw1] (*Figure 3A*). As with the Jumble mutant, we were able to rescue rosette formation in a portion of the population by transfecting cells with either a *couscous-mTFP* or *mTFP-couscous* gene fusion under the *efl* promoter (*Figure 3C,D*), thereby increasing confidence in the mapping results.

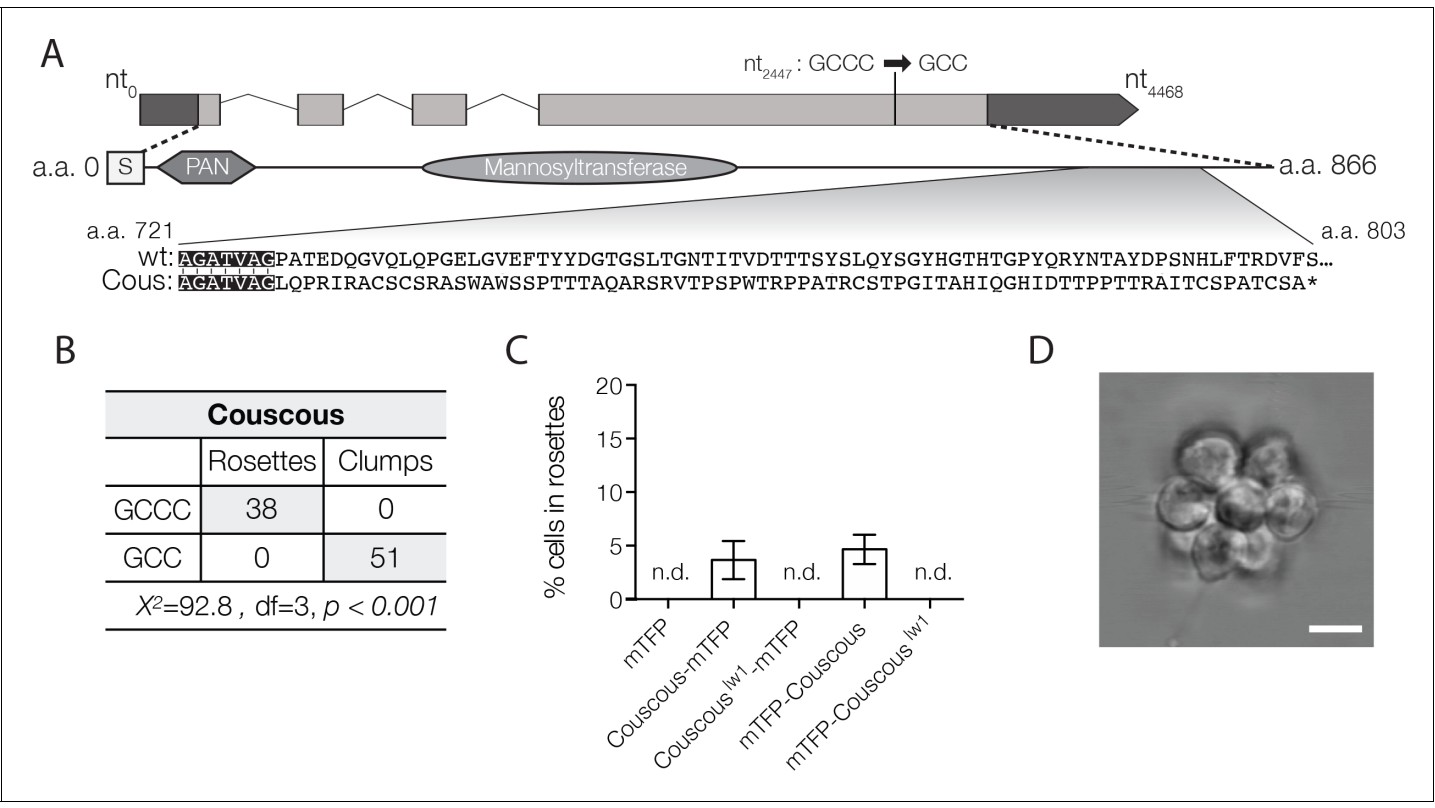

**Figure 3.** Couscous maps to a predicted mannosyltransferase with a PAN/Apple domain. (A) Couscous has a predicted signal sequence (S), a PAN/Apple domain (PAN), and a mannosyltransferase domain. The causative lesion is a 1-base pair deletion at nucleotide position 2447 that causes a frameshift at amino acid 728, resulting in 75 amino acids that do not align between the wild type and mutant (Cous) sequences, and an early stop codon (*) at amino acid 803. (B) Independent backcrosses of two individual mutant F1 progeny to the Mapping Strain yielded 38 rosette-forming F2 isolates with the wild type GCCC allele and 51 clumpy F2 isolates with the *couscous^lw1* GCC allele. The inheritance significantly deviated from expected Mendelian inheritance of unlinked traits and confirmed the tight linkage between the *couscous^lw1* allele and the clumpy, rosetteless phenotype. $X^2$ = Chi squared value, d.f. = degrees of freedom. (C, D) Rosette formation in Couscous mutant cells can be rescued by transgenic expression of *couscous-mTFP* or *mTFP-couscous*, but not *couscous^lw1-mTFP*, *mTFP-couscous^lw1*, or *mTFP* alone. RIFs were added immediately after transfection and 40 µg/ml puromycin was added 24 hr post-transfection to select for positive transformants. (C) Rosette development (mean ± SD) was measured as the % of cells in rosettes 72 hr after transfection and treatment with RIFs. n.d. = no detected rosettes. (n = 200 cells counted from each of 3 technical replicates; two biological replicates). (D) Rosettes transgenically complemented with *couscous-mTFP* in the Couscous mutant appeared phenotypically wild type. Representative rosette shown. Scale bar = 5 µm.

DOI: https://doi.org/10.7554/eLife.41482.013

The following figure supplement is available for figure 3:

**Figure supplement 1.** Couscous homology and localization.
DOI: https://doi.org/10.7554/eLife.41482.014

The predicted Couscous amino acid sequence contains a specific type of glycosyltransferase domain, an alpha-mannosyltransferase domain, that transfers activated mannose onto the outer chain of core N-linked polysaccharides and O-linked mannotriose (*Strahl-Bolsinger et al., 1999*). The predicted mannosyltransferase domain shares 28% and 35% amino acid sequence identity to alpha 1–2 mannosyltransferase (MNN2) proteins in *Saccharomyces cerevisiae* and *Candida albicans*, respectively, including the conserved DXD motif found in many families of glycosyltransferases (*Wiggins and Munro, 1998*) (*Figure 3—figure supplement 1A*). MNN2 proteins catalyze the addition of the first branch of mannose-containing oligosaccharides found on cell wall proteins (*Rayner and Munro, 1998*) and proper MNN2 activity is required for flocculation, or non-mating aggregation, in *S. cerevisiae* (*Stratford, 1992*). In addition to the mannosyltransferase domain, Couscous is predicted to have a PAN/Apple domain composed of a conserved core of three disulfide bridges (*Ho et al., 1998*; *Tordai et al., 1999*). PAN/Apple domains are broadly distributed

among eukaryotes, including animals, where they mediate protein-protein and protein-carbohydrate interactions, often on the extracellular surface of the cell (*Ho et al., 1998*; *Tordai et al., 1999*).

In wild type cells transfected with a *couscous-mWasabi* transgene under the *efl* promoter, Couscous was found in puncta scattered throughout the cytosol, collar and cell membrane (*Figure 3—figure supplement 1B,C*). While Couscous-mWasabi was clearly not localized to the Golgi, the puncta may co-localize with the ER, where glycosyltransferases are also known to function (*El-Battari, 2006*; *Tu and Banfield, 2010*). However, despite attempting to co-transfect cells with *couscous-mWasabi* and a marker of the ER (mCherry fused to a C-terminal HDEL ER retention signal sequence (*Booth et al., 2018*)), we were unable to detect any cells expressing both gene fusions. In addition, it is possible that the fusion of Couscous to a fluorescent protein or its overexpression interfered with its proper localization in *S. rosetta*. Therefore, we are currently uncertain about the subcellular localization of Couscous protein.

## Jumble and Couscous mutants lack proper sugar modifications at the basal pole

Because both Jumble and Couscous have mutations in predicted glycosyltransferases, we hypothesized that the abundance or distribution of cell surface sugars, called glycans, on Jumble and Couscous mutant cells might be altered. To investigate the distribution of cell surface glycans, we stained live *S. rosetta* with diverse fluorescently labelled sugar-binding lectins. Of the 22 lectins tested, 21 either did not recognize *S. rosetta* or had the same staining pattern in wild type, Jumble and Couscous cells (*Supplementary file - Table S5*).

The remaining lectin, jacalin, bound to the apical and basal poles of wild type cells (*Figure 4A,B, B′*). Jacalin also brightly stained the ECM filling the center of rosettes in a pattern reminiscent of the Rosetteless C-type lectin (*Levin et al., 2014*) (*Figure 4A,B′*), although the two were not imaged simultaneously because jacalin does not bind after cell fixation and labelled Rosetteless antibodies accumulate strongly in the food vacuoles of live cells. In contrast with wild type cells, the basal patch of jacalin staining was absent or significantly diminished in Couscous and Jumble mutants, both in the presence and absence of RIFs (*Figure 4C–F*). Interestingly, the apical patch of jacalin staining in mutant cells appeared similar to wild type cells. This may explain the lack of a clear difference in bands detected with jacalin by western blot between wild type and mutants whole cell lysates (*Figure 4—figure supplement 1*). Transformation of Jumble cells with *mTFP-jumble* not only rescued rosette development (*Figure 2C,D*), but also restored the wild type glycosylation pattern, as revealed by jacalin staining in the center of complemented rosettes (*Figure 4—figure supplement 2*). The same was true for Couscous cells, in which transformation with *couscous-mTFP* rescued both rosette development and the wild type glycosylation pattern (*Figure 3C,D*; *Figure 4—figure supplement 2*). Thus, the glycosylation defects in Jumble and Couscous mutant cells were directly linked to the genetic lesions in *jumble* and *couscous,* respectively.

The loss of basal jacalin staining in Jumble and Couscous mutants indicated that *jumble*[lw1] and *couscous*[lw1] either disrupt proper trafficking of sugar-modified molecules to the basal pole of cells or alter the glycosylation events themselves. Thus, we examined whether the basal secretion of Rosetteless protein was disrupted in the mutant strains. In both Jumble and Couscous cells, Rosetteless protein properly localized to the basal pole, but its expression did not increase nor was it secreted upon treatment with RIFs, as normally occurs in wild type cells (*Figure 4—figure supplement 3*). Because Rosetteless is required for rosette development, this failure to properly upregulate and secrete Rosetteless might contribute to the rosette defect phenotype in Jumble and Couscous cells.

## Discussion

Of the 16 rosette defect mutants isolated in *Levin et al. (2014)* and in this study, almost half (7) also display a mild to severe clumping phenotype. This suggests that mechanisms for preventing promiscuous adhesion among wild type cells can be easily disrupted. We found that the clumping phenotype results from promiscuous adhesion of mutant cells to other mutant or wild type cells rather than from incomplete cytokinesis. A recent study revealed that the bacterium *Vibrio fischeri* induces *S. rosetta* to form swarms of cells, visually similar to the mutant clumps, as part of their mating behavior (*Woznica et al., 2017*). However, it seems unlikely that the clumping Class C mutants is

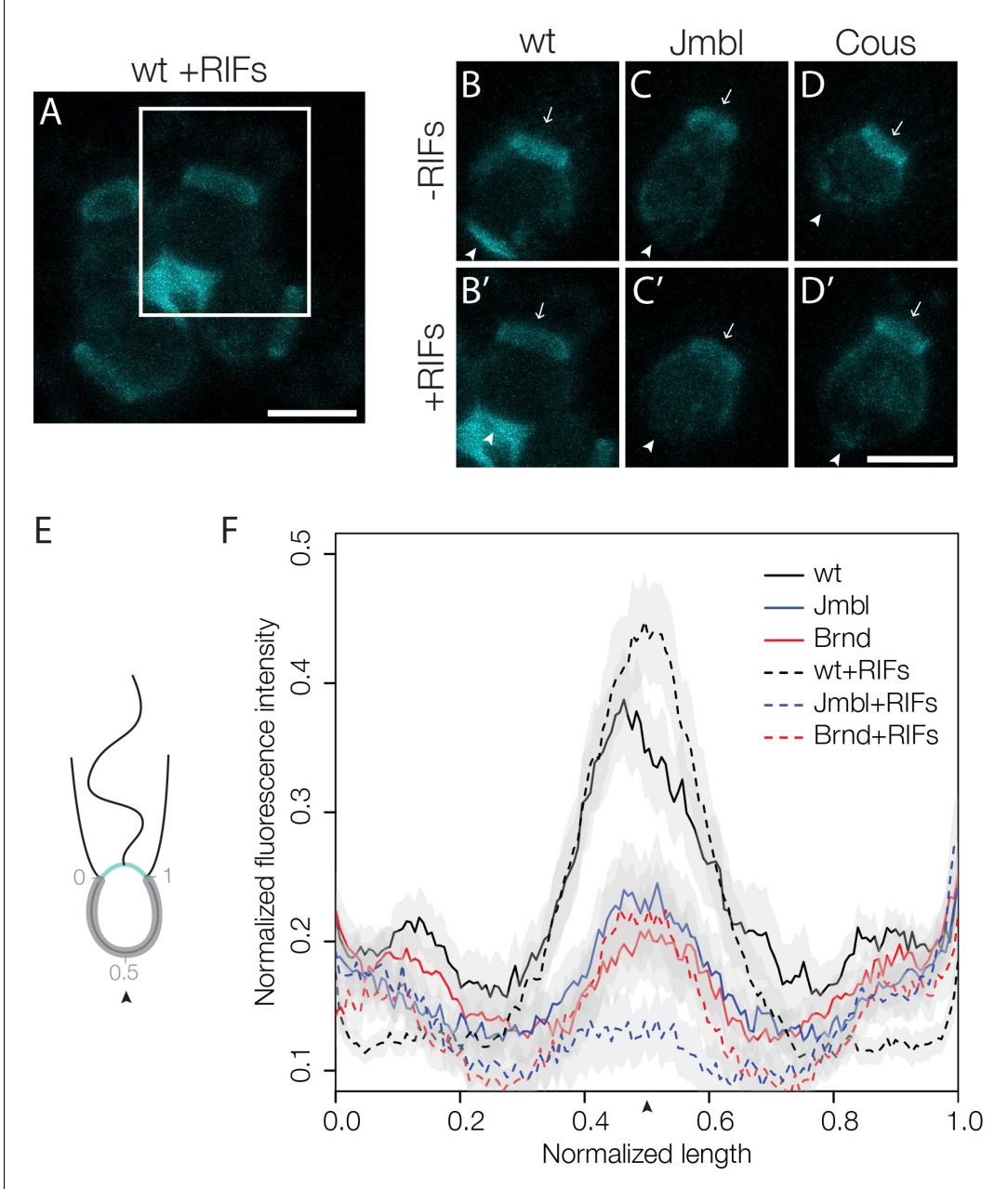

**Figure 4.** Disruption of basal glycosylation patterns in Jumble and Couscous mutants. FITC-labelled jacalin binds the apical and basal poles of wild type single cells (**B**) and becomes enriched in the ECM in the center of rosettes (**A, B'** boxed region from **A**). Although FITC-jacalin staining appeared normal at the apical poles of Jumble (**C**) and Couscous (**D**) mutant cells, FITC-jacalin staining at the basal poles of cells was undetectable in cells grown either in the absence (-RIFs; **C, D**) or presence (+RIFs; **C', D'**) of RIFs. Arrows mark the apical pole and arrowheads mark the basal pole. (**E**) Cartoon depicts how jacalin fluorescence was measured. Starting with micrographs of FITC-jacalin stained cells, a line was drawn tracing from one edge of the collar around the cell body to the other edge of the collar, and the underlying fluorescent signal was normalized for cell size and background intensity. (**F**) The average normalized fluorescence intensity of jacalin measured in at least 59 cells for each condition was graphed against the normalized length of the cell body (n = 2 biological replicates). Jumble and Couscous -/+RIFs have reduced jacalin binding at the basal pole compared to wild type -/+RIFs. Gray shadows indicate 95% confidence intervals. Scale bars = 5 μm.

DOI: https://doi.org/10.7554/eLife.41482.015

The following figure supplements are available for figure 4:

**Figure supplement 1.** Jacalin Western blot in cell lysates.

DOI: https://doi.org/10.7554/eLife.41482.016

**Figure supplement 2.** Transgenic rescue restores jacalin staining at the center of complemented rosettes.

*Figure 4 continued on next page*

*Figure 4 continued*

DOI: https://doi.org/10.7554/eLife.41482.017

**Figure supplement 3.** Rosetteless staining in wild type and mutant cells.

DOI: https://doi.org/10.7554/eLife.41482.018

related to swarming; the cell fusion and subsequent settling of diploid cells characteristic of *V. fischeri*-induced mating have not been observed in the class C mutants cultured without *V. fischeri*.

For both Jumble and Couscous, the causative mutations mapped to predicted glycosyltransferase genes. Consistent with its role as a glycosyltransferase, Jumble localized to the Golgi apparatus, but Couscous appeared to localize in cytoplasmic puncta and to the cell membrane. We predict that the mutations in predicted glycosyltransferases are loss of function alleles, given that transfection of mutant *S. rosetta* with the wild type alleles was sufficient to complement each of the mutations. While we have not uncovered the target(s) of the predicted glycotransferases or the exact nature of the interplay between the two phenotypes, disruption of the glycocalyx at the basal pole of both Jumble and Couscous mutant cells (*Figure 4*) hints that the regulation of ECM could play a role in preventing clumping and in promoting proper rosette development.

One possible explanation for the clumping phenotype is that *jumble* and *couscous* are required to regulate the activity of cell surface adhesion molecules and receptors. Glycosylation regulates the activities of two key adhesion proteins in animals: integrins that regulate ECM adhesion, and cadherins that, among their various roles in cell signaling and animal development, bind other cadherins to form cell-cell adhesions called adherens junctions (*Larsen et al., 2017*; *Zhao et al., 2008*). Cadherin activity can be either positively or negatively regulated by glycosyltransferases. For example, epithelial cadherin (E-cadherin) is modified by N-acetylglucosaminyltransferase III (GnT-III) whose activity leads to increased cell adhesion and N-acetylglucosaminyltransferase V (GnT-V) whose activity leads to decreased cell adhesion (*Carvalho et al., 2016*; *Granovsky et al., 2000*). GnT-V knockdown enhances cell-cell adhesion mediated by E-cadherin and the related N-cadherin (*Carvalho et al., 2016*; *Guo et al., 2009*). The inactivation of E-cadherin, including through over- or under- expression of GnT-V or GnT-III, is considered to be a hallmark of epithelial cancers (*Hirohashi and Kanai, 2003*). *S. rosetta* expresses 29 different cadherins (*Nichols et al., 2012*) and it is possible that mutations to *jumble* and *couscous* disrupt regulatory glycosylation of a cell adhesion molecules like cadherins.

Another possibility is that *jumble* and *couscous* add a protective sugar layer to the cell surface and loss of glycosyltransferase activity reveals underlying sticky surfaces. If *jumble* and *couscous* add branches to existing sugar modifications, their loss of function could expose new sugar moieties at the cell surface that act as ligands for lectins that aggregate cells. Lectins mediate cell aggregation in diverse organisms (*Colin Hughes, 1992*). For example, sponges, such as *Geodia cydonium,* can be disaggregated into single cells and then reaggregated through lectin binding of a post-translational sugar modification (*Müller et al., 1979*). In *S. cerevisiae*, the mannosyltransferase MNN2 adds mannose structures to the cell wall that are recognized by aggregating lectins and MNN2 is required for proper flocculation (*Rayner and Munro, 1998*; *Stratford, 1992*). Exposing new sugars on the cell surface in Jumble and Couscous could lead to spurious aggregation, potentially by lectins or other sugar binding proteins.

It is somewhat more difficult to infer how increased clumping in single cells might interfere with rosette development. One possibility is that the disruption of ECM glycosylation that we hypothesize might promote clumping may also prevent the proper maturation of the ECM needed for rosette development (*Figure 5*). A prior study showed that only *S. rosetta* cells recognized with the lectin wheat germ agglutinin (WGA) are competent to form rosettes, which suggests that glycosylation might be necessary for rosette development (*Dayel et al., 2011*). While WGA staining does not appear to be perturbed in Jumble and Couscous (*Supplementary file - Table S5*), jacalin staining at the basal pole appears severely reduced or abolished compared to wild type. Jacalin staining was enriched in the center of wild type rosettes in a pattern reminiscent of Rosetteless, which is required for rosette development (*Levin et al., 2014*). Intriguingly, in Jumble and Couscous, Rosetteless localized to the correct pole, but did not become enriched upon rosette induction, indicating that the ECM did not properly mature. Rosetteless has mucin-like Ser/Thr repeats that are predicted sites of heavy glycosylation and two C-type lectin domains that would be expected to bind to sugar

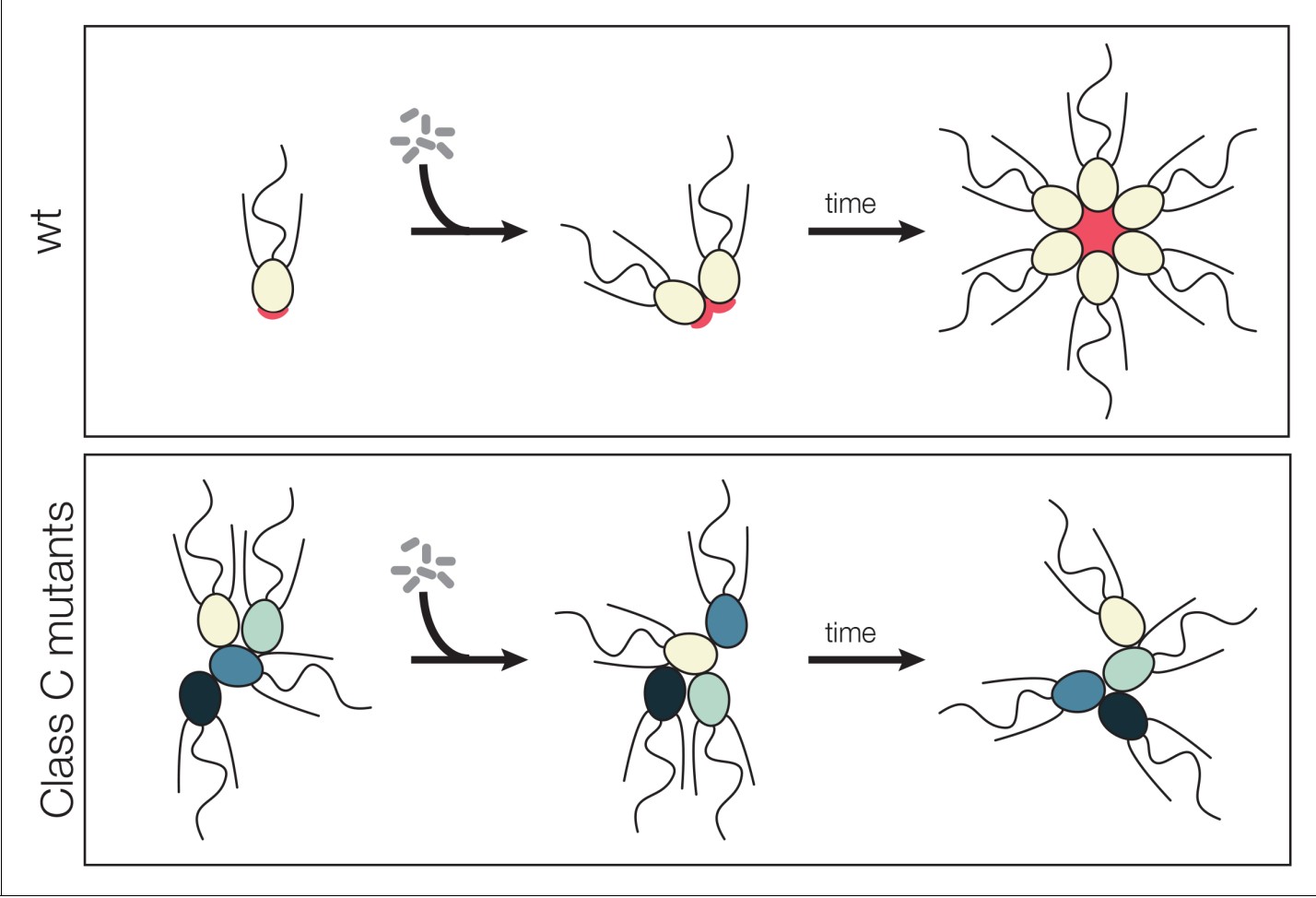

**Figure 5.** Model for promiscuous clumping in rosette defective Class C mutants. Wild type *S. rosetta* has a glycosylated basal patch of ECM (red) as marked by the lectin jacalin that becomes enriched during the course of rosette formation. The Rosetteless protein, required for rosette formation and speculated to play a structural role in holding rosettes together, localizes to the same location on the basal pole of cells and becomes similarly enriched as rosette form. Mutants lack the glycosylated basal patch of jacalin staining. The altered cell surface could lead to clumping, either through mis-regulation of cell adhesion molecules or exposure of a normally masked adhesive cell surface. The same alteration that allows clumping of Class C mutants also prevents rosette development, perhaps by disrupting glycan modification on the Rosetteless protein or one of its interaction partners.
DOI: https://doi.org/10.7554/eLife.41482.019

moieties (*Levin et al., 2014*). Therefore, it is possible that Rosetteless might be regulated either through direct glycosylation or through the glycosylation of potential binding partners by Jumble and Couscous.

The clumping, rosette defect mutants underscore the differences between cell aggregation and a regulated clonal developmental program, such as animal embryogenesis or choanoflagellate rosette development. Importantly, the multicellularity of all extant animals arises through cell division rather than cell aggregation, suggesting that the suppression of cell aggregation in favor of clonal development was a prerequisite for animal origins. Our results raise the possibility that glycosylation and the regulation of the ECM suppressed cell aggregation while stabilizing obligate clonal multicellularity on the animal stem lineage. Glycosylation remains an important regulator of tissue organization in modern animals (*Zhang et al., 2008*). Interestingly, cancer suppression is thought to have been important for ensuring organismal integrity in multicellular animals (*Aktipis et al., 2015*) and disruption of glycosylation is often implicated in metastatic cancers (*Pinho and Reis, 2015*). Understanding the molecular mechanisms that prevent spurious aggregation in *S. rosetta* may provide new insights into the mechanisms that ensured cell cooperativity in stem animals while also revealing cancer vulnerabilities in modern animals.

# Materials and methods

## Key resources table

| Reagent type (species) or resource | Designation | Source or reference | Identifiers | Additional information |
|---|---|---|---|---|
| Gene (*Salpingoeca rosetta*) | *jumble* | NA | GenBank accession EGD72416/NCBI accession XM_004998928; | |
| Gene (*S. rosetta*) | *couscous* | NA | GenBank accession EGD77026/NCBI accession XM_004990809 | |
| Strain, strain background (*S. rosetta*) | wt | PMID: 24139741 | ATCC PRA-390; accession number SRX365844 | |
| Strain, strain background (*S. rosetta*) | Mapping Strain | PMID: 24139741 | accession number SRX365839 | |
| Strain, strain background (*S. rosetta*) | Jumble | PMID: 25299189 | accession number SRR7866767 | |
| Strain, strain background (*S. rosetta*) | Couscous | PMID: 25299189 | accession number SRR7866768 | Previously named Branched |
| Strain, strain background (*S. rosetta*) | Seafoam | PMID: 25299189 | accession number SRR8263910 | |
| Strain, strain background (*S. rosetta*) | Soapsuds | PMID: 25299189 | accession number SRR8263909 | |
| Strain, strain background (*Algoriphagus macihipongenesis*) | *Algoriphagus macihipongenesis* | PMID: 22368173 | ATCC BAA-2233 | |
| Strain, strain background (*Echinicola pacifica*) | *Echinicola pacifica* | PMID: 16627637 | DSM 19836 | |
| Strain, strain background (*Vibrio fishceri*) | *Vibrio fishceri* ES114 | PMID: 15703294 | ATCC 700601 | |
| Antibody | anti-Rosetteless | PMID: 25299189 | | (1:400) |
| Recombinant DNA reagent | mCherry plasma membrane marker | PMID: 30281390 | RRID:Addgene_109094; Addgene ID NK624 | |
| Recombinant DNA reagent | mCherry ER marker | PMID: 30281390 | RRID:Addgene_109096; Addgene ID NK644 | |
| Recombinant DNA reagent | *pEFl5'-Actin3'::jumble-mWasabi* | this paper | Addgene ID NK690 | pUC19 backbone with 5' *S. rosetta* elongation factor L (*efl*) promoter, j*umble*, *mWasabi*, and 3' UTR from actin; assembled by Gibson assembly |
| Recombinant DNA reagent | *pEFl5'-Actin3'::jumble$^{lw1}$-mWasabi* | this paper | Addgene ID NK691 | pUC19 backbone with 5' *S. rosetta* elongation factor L (*efl*) promoter, j*umble*$^{lw1}$, *mWasabi*, and 3' UTR from actin; assembled by Gibson assembly |

*Continued on next page*

*Continued*

| Reagent type (species) or resource | Designation | Source or reference | Identifiers | Additional information |
|---|---|---|---|---|
| Recombinant DNA reagent | *pEFl5'-Actin3':: couscous-mWasabi* | this paper | Addgene ID NK692 | pUC19 backbone with 5' *S. rosetta* elongation factor L (*efl*) promoter, *couscous*, *mWasabi*, and 3' UTR from actin; assembled by Gibson assembly |
| Recombinant DNA reagent | *pEFl5'-Actin3':: pac-P2A-mTFP* | this paper | Addgene ID NK676 | pUC19 backbone with 5' *S. rosetta* elongation factor L (*efl*) promoter, *S. rosetta* codon optimized puromycin resistance gene (*pac*), *mTFP*, and 3' UTR from actin; assembled by Gibson assembly |
| Recombinant DNA reagent | *pEFl5'-Actin3'::pac -P2A-jumble-mTFP* | this paper | Addgene ID NK694 | Parent vector: *pEFl5'-Actin3'::pac-P2A-mTFP*; *jumble* inserted using Gibson assembly |
| Recombinant DNA reagent | *pEFl5'-Actin3'::pac -P2A-jumble$^{lw1}$-mTFP* | this paper | Addgene ID NK695 | Parent vector: *pEFl5'-Actin3'::pac-P2A-mTFP*; *jumble$^{lw1}$* inserted using Gibson assembly |
| Recombinant DNA reagent | *pEFl5'-Actin3'::pac -P2A-mTFP-jumble* | this paper | Addgene ID NK696 | Parent vector: *pEFl5'-Actin3'::pac-P2A-mTFP*; *jumble* inserted using Gibson assembly |
| Recombinant DNA reagent | *pEFl5'-Actin3'::pac -P2A-mTFP-jumble$^{lw1}$* | this paper | Addgene ID NK697 | Parent vector: *pEFl5'-Actin3'::pac-P2A-mTFP*; *jumble$^{lw1}$* inserted using Gibson assembly |
| Recombinant DNA reagent | *pEFl5'-Actin3'::pac -P2A-couscous-mTFP,* | this paper | Addgene ID NK698 | Parent vector: *pEFl5'-Actin3'::pac-P2A-mTFP*; *couscous* inserted using Gibson assembly |
| Recombinant DNA reagent | *pEFl5'-Actin3'::pac -P2A-couscous$^{lw1}$-mTFP* | this paper | Addgene ID NK699 | Parent vector: *pEFl5'-Actin3'::pac-P2A-mTFP*; *couscous$^{lw1}$* inserted using Gibson assembly |
| Recombinant DNA reagent | *pEFl5'-Actin3'::pac -P2A-mTFP-couscous* | this paper | Addgene ID NK700 | Parent vector: *pEFl5'-Actin3':: pac-P2A-mTFP*; *couscous* inserted using Gibson assembly |
| Recombinant DNA reagent | *pEFl5'-Actin3'::pac- P2A-mTFP-couscous$^{lw1}$* | this paper | Addgene ID NK701 | Parent vector: *pEFl5'-Actin3':: pac-P2A-mTFP*; *couscous$^{lw1}$* inserted using Gibson assembly |
| Other | FITC-labelled jacalin | Vector Labs | RRID:AB_2336460; Vector Labs: Cat. No.FLK-4100 | (1:400) |
| Other | biotinylated jacalin | Vector Labs | RRID:AB_2336541; Vector Labs: Cat. No. B-1155 | |
| Other | Streptavidin Alexa Fluor 647 conjugate | Thermo Fisher Scientific | Thermo Fisher Scientific: Cat. No. 32357 | |

## Media preparation, strains, and cell culture

Unenriched artificial seawater (ASW), AK artificial seawater (AK), cereal grass media (CG), and high nutrient (HN) media were prepared as described previously (*Booth et al., 2018*; *Levin et al., 2014*; *Levin and King, 2013*). The wild type strain, from which each mutant was generated, was the described strain SrEpac (ATCC PRA-390; accession number SRX365844) in which *S. rosetta* is co-cultured monoxenically with the prey bacterium *Echinicola pacifica* (DSM 19836, *Levin et al., 2014*; *Levin and King, 2013*; *Nedashkovskaya et al., 2006*). Seafoam, Soapsuds, and Couscous (previously named Branched) were generated through X-ray mutagenesis and Jumble was generated by EMS mutagenesis as documented in *Levin et al. (2014)*. In *Levin et al. (2014)*, Branched/Couscous was not thoroughly characterized and was named based on the hypothesis that the clumps formed through cell divisions that resulted in unusually branched chain colonies. (Wild type chain colonies are primarily linear, with rare branches.) Our thorough characterization of the mutant in this study revealed that the clumps form through aggregation and not through branching cell divisions. In order for the mutant name to better reflect the phenotype, we renamed it Couscous. For routine culturing, wild type and mutant cultures were diluted 1:10 every 2–3 days in 5% (v/v) HN media in ASW. The Mapping Strain, (previously called Isolate B in *Levin et al., 2014*) used for mapping crosses (accession number SRX363839) was grown in the presence of rosette-inducing *A. machipongonensis* bacteria (ATCC BAA-2233). The Mapping Strain was maintained in 25% (v/v) CG media diluted in ASW and passaged 1:10 every 2–3 days. For transfection of *S. rosetta*, cells were maintained in 5% (v/v) HN media in AK (*Booth et al., 2018*). Rosette formation initially was assayed using both live *A. machipongonensis* and *A. machipongonensis* outer membrane vesicles (OMVs) prepared as in *Woznica et al. (2016)*. For each strain tested, both methods of rosette induction resulted in similarly low/non-existent percentages of cells in rosettes and visually similar clumps for Class C mutants (*Supplementary file - Table S1*). Therefore, unless stated otherwise, rosette induction was performed with *A. machipongonensis* OMVs and referred to here as rosette inducing factors (RIFs).

## Imaging and quantifying rosette phenotypes

To image rosette phenotypes (*Figure 1A*), cells were plated at a density of $1 \times 10^4$ cells/ml in 3 ml 5% (v/v) HN media in ASW either with or without *Algoriphagus* RIFs. Cultures were imaged after 48 hr of rosette induction in 8-well glass bottom dishes (Ibidi 15 µ-Slide eight well Cat. No. 80826) that were coated with 0.1 mg/ml poly-D-lysine (Sigma) for 15 min and washed 3 times with water to remove excess poly-D-lysine. For imaging wild type and mutant cultures in the presence and absence of RIFs (*Figure 1A* top two panels), 200 µl of cells were plated with a wide bore pipette tip for minimal disruption and allowed to settle for 5 min. For images of vortexed cells (*Figure 1A* bottom panel), 200 µl of cells were vortexed for 15 s before plating and imaged within 10 min of plating to prevent re-clumping. Cells were imaged live by differential interference contrast microscopy using a Zeiss Axio Observer.Z1/7 Widefield microscope with a Hammatsu Orca-Flash 4.0 LT CMOS Digital Camera and a 63x/NA1.40 Plan-Apochromatic oil immersion lens with 1.6X optivar setting.

To quantify rosette induction (*Figure 1B*), cells were plated at a density of $1 \times 10^4$ cells/ml in 3 ml 5% (v/v) HN media in ASW with RIFs. After 48 hr, an aliquot of cells was vortexed vigorously for 15 s and fixed with formaldehyde. To determine the percentage of cells in rosettes, the relative number of single cells and cells within rosettes were scored using a hemocytometer. Rosettes were counted as a group of 3 or more cells with organized polarity relative to a central focus after exposure to vortexing.

## Imaging and quantification of cell clumping

Clumps were quantified using a modified protocol from *Woznica et al. (2017)* (*Figure 1C*; *Figure 1—figure supplement 3*). Mutant cells, and to some extent wild type cells, will adhere to glass. Therefore, to prevent cells from simply sticking to the bottom of the 8-well glass bottom dishes (Ibidi 15 µ-Slide eight well Cat. No. 80826), the dishes were coated with 1% BSA for 1 hr and washed 3 times with water to remove any residual BSA. Importantly, the addition of BSA to the imaging dishes did not cause wild type cells to stick to the bottom of the dishes or to each other. Cells were diluted to $5 \times 10^5$ cells/ml, vortexed for 15 s to break apart any pre-formed clumps and plated in the BSA pre-treated dishes. For quantification, DIC images were taken using a Zeiss Axio Observer.Z1/7

Widefield microscope with a Hammatsu Orca-Flash 4.0 LT CMOS Digital Camera and a 20x objective. Images were collected for each strain from 10 distinct locations throughout the well.

Images were batch processed in ImageJ for consistency. To accurately segment the phase bright cells and limit signal from the phase dark bacteria the following commands were applied with default settings: 'Smooth' (to reduce background bacterial signal), 'Find Edges' (to highlight the phase-bright choanoflagellate cells), 'Despeckle' (to remove noise), 'Make Binary' (to convert to black and white), 'Dilate' (to expand to smooth jagged edges from segmentation), 'Erode' (to return to the same size as before dilate), and 'Fill Holes' (to fill any remaining small holes). Finally, images were analyzed with the 'Analyze Particles' command to calculate the area of the clump and only particles larger than 20 $\mu m^2$ were kept to filter out any remaining bacterial signal. Cell equivalents/clump (*Figure 1C* and *Figure 1—figure supplement 3*, right y axis) were calculated by dividing the area of the clump by the area of a representative individual cell (as approximated by averaging the area of the wild type cells). Data are presented as violin boxplots, showing the median cell number (horizontal line), interquartile range (white box), and range excluding outliers (vertical line). A minimum of 630 clumps from two biological replicates were measured for each condition.

To examine whether cell division was required for clump formation, we used aphidicolin to block cell division (*Figure 1—figure supplement 1*). Cells in vortexed wild type, Jumble, and Couscous cultures were counted and diluted to $1 \times 10^5$ cells/ml in 5% (v/v) HN media in AK. For each strain, either 250 µM aphicidolin, an equal volume of a DMSO control, or no additional control were added to each condition. After 24 hr, DIC images were taken using a Zeiss Axio Observer.Z1/7 Widefield microscope with a Hammatsu Orca-Flash 4.0 LT CMOS Digital Camera and a 40x air objective.

## Performing mapping crosses

Mapping crosses for each mutant strain (Seafoam, Soapsuds, Jumble, and Couscous) with Mapping Strain (previously described as Isolate B) were attempted using both methods previously shown to induce mating in *S. rosetta*: nutrient limitation for 11 days and addition of 2.5–5% *Vibrio fischeri* (ATCC 700601) conditioned media (*Levin and King, 2013*; *Woznica et al., 2017*). Both methods were effective at inducing mating for all attempted crosses; here, we report which method was used to generate data for each individual cross. Cells induced to mate were plated by limiting dilution to isolate diploid clones. Clonal isolates were allowed to grow for 5–7 days and screened for populations of thecate cells, as these are the only documented diploid cell type (*Levin et al., 2014*; *Woznica et al., 2017*). From each population of thecate cells, we extracted DNA from 75 µl of cells by scraping cells from the plate, harvesting and pelleting the cells, resuspending in 10 µl of base solution (25 mM NaOH, 2 mM EDTA), transferring samples into PCR plates, boiling at 100°C for 20 min, followed by cooling at 4°C for 5 min, and then adding 10 µl Tris solution (40 mM Tris-HCl, pH 7.5). We used 2 µl of this sample as the DNA template for each genotyping reaction. We identified heterozygous strains through genotyping by PCR at a single microsatellite genotyping marker at position 577,135 on supercontig 1 (Forward primer: GACAGGGCAAAACAGACAGA and Reverse primer: CCATCCACGTTCATTCTCCT) that distinguishes a 25 bp deletion in the Mapping Strain (199 bp) from the strain used to generate the mutants (217 bp). Isolates containing PCR products of both sizes were inferred to be diploid. Meiosis was induced by rapid passaging every day in CG medium.

For both Seafoam and Soapsuds, we were able to generate putative outcrossed diploids by crossing to the Mapping Strain based on the genotyping marker on supercontig 1, but we only could only clonally isolate populations of F1 haploids with rosettes and never isolated any F1 haploids with the clumpy, rosetteless phenotype. Whole genome resequencing of Seafoam and Soapsuds revealed no mutations at the *rosetteless*, *jumble*, or *couscous* loci. Seafoam and Soapsuds have 17 and 34 predicted nonsense or missense mutations, respectively, in coding sequences and additional mutations in non-coding portions of the genome. Of the lesions in Seafoam and Soapsuds, none were detected in genes encoding a predicted glycosyltransferase, lectin, or related gene family. Without being able to do mapping crosses, it was not possible to identify the causative mutations from Seafoam or Soapsuds.

For the successful cross of Jumble to the Mapping Strain, we induced mating by starvation using the approach of *Levin and King (2013)*. First, we started with rapidly growing, regularly passaged strains, pelleted $2 \times 10^6$ cells/ml of each strain together and resuspended in 10 ml of ASW lacking any added nutrients. After 11 days of starvation in ASW, we pelleted all cells (presumably including diploid cells resulting from mating) and resuspended in 100% CG media to recover any diploids.

After 3 days of recovery, we isolated clones by limiting dilution in 10% (v/v) CG media in ASW. The probability of clonal isolation in this step was 0.91–0.93 (calculated using the Poisson distribution and the number of choanoflagellate-free wells per plate; *Levin and King, 2013*). Three clonally isolated heterozygous populations, each containing almost exclusively thecate cells, were identified through genotyping by PCR at a supercontig 1 microsatellite as described above. To induce meiosis, heterozygotes were diluted 1:2 in 25% (v/v) CG media in ASW every 1–2 days for 8 days. As soon as rosettes and swimming cells were observed, we repeated the serial dilution to isolate clones (probability of clonal isolation 0.85–0.98). We collected any clonally isolated populations that formed rosettes or clumps and ignored any wells containing thecate cells assuming that these represented diploid cells that had not undergone meiosis. 56% of all non-thecate isolates displayed the cell clumping phenotype and 44% of all non-thecate isolates were capable of forming rosettes, consistent with Mendelian segregation of a single locus ($\chi^2$=1.162, df = 1, p=0.28). Isolates were genotyped with the marker on supercontig 1 to ensure that independent assortment of the genotype and the phenotype indeed occurred. In total, 30 clumpy F1s were collected for bulk segregation analysis.

For the successful Couscous cross, we induced mating using *V. fisheri* conditioned media using the approach of *Woznica et al. (2017)*. A mixture of $1 \times 10^6$ Couscous and Mapping Strain cells at stationary growth were pelleted and resuspended in 5% (v/v) *V. fischeri* conditioned media in ASW. After 24 hr, the cells were pelleted, resuspended in 5% (v/v) HN media in ASW, and allowed to recover for 24 hr. We then isolated clones by limiting dilution in 10% (v/v) CG media in ASW. The probability of clonal isolation in this step was between 0.97–0.98. We extracted DNA as described above and identified heterozygous clones through genotyping by PCR at a single microsatellite genotyping marker on supercontig 1. Four clonally isolated heterozygous populations, containing almost exclusively thecate cells, were identified. To induce meiosis, heterozygotes were passaged 1:2 in 25% (v/v) CG media in ASW every 1–2 days for 8 days. As soon as rosettes and swimming cells were observed, we repeated clonal isolation (probability of clonal isolation 0.78–0.97). We collected any clonally isolated populations that formed rosettes or clumps and ignored any wells containing thecate cells assuming that these represented diploid cells that had not undergone meiosis. Only 14.6% of non-thecate isolates were clumps; this deviation from a Mendelian ratio ($\chi^2$=225.63, df = 1, p<5.34$^{-51}$) may indicate a potential fitness defect of the mutant phenotype. Isolates were genotyped with the marker on supercontig 1 to ensure that independent assortment indeed occurred. In total, 22 clumpy F1s were collected for bulk segregant analysis.

## Whole genome sequencing

Jumble, Couscous, Seafoam, and Soapsuds were whole genome sequenced individually to identify the mutation(s) carried in each strain. To do this, Jumble, Couscous, Seafoam, and Soapsuds cells were grown to stationary phase in 500 ml of 5% (v/v) HN media in ASW. To generate pooled genomic DNA for bulk segregant analysis, we grew up $5 \times 10^6$ cells of each of the 38 F1s with the rosetteless phenotype from the Rosetteless ×Mapping Strain cross (*Levin et al., 2014*), $5 \times 10^6$ cells of each of the 30 F1s with the clumpy phenotype from the Jumble × Mapping Strain cross, and $5 \times 10^6$ cells of each of the 22 F1s with the clumpy phenotype from the Couscous × Mapping Strain cross. For each cross, the F1 cells were pelleted, frozen, and combined during lysis for DNA extraction. For all samples, we performed a phenol-chloroform DNA extraction and used a CsCl gradient to separate *S. rosetta* DNA from contaminating *E. pacifica* DNA by GC content (*King et al., 2008*).

Multiplexed, 100 bp paired-end libraries were prepared and sequenced on an Illumina HiSeq 2000 for the Jumble, Couscous, Seafoam, and Soapsuds mutant DNA alone. Multiplexed, 150 bp paired-end libraries were prepared and sequenced on an Illumina HiSeq 2500 for the Rosetteless ×Mapping Strain cross and the Jumble x Mapping Strain cross pooled DNA. For the Couscous × Mapping Strain cross DNA, a multiplexed, 300 bp paired-end library was prepared and sequenced on an Illumina MiSeq. Raw reads are available at the NCBI Short Read Archive with the BioProject identifier PRJNA490902. BioSample and SRA accession numbers are as follows: Jumble mutant-SAMN10061445 and SRR7866767, Couscous mutant-SAMN10061446 and SRR7866768, Seafoam mutant-SAMN10501893 and SRR8263910, Soapsuds mutant- SAMN10501894 and SRR8263909, Rosetteless × Mapping Strain cross-SAMN10061447 and SRR7866769, Jumble × Mapping Strain cross-SAMN10061448 and SRR7866770, and Couscous × Mapping Strain cross- SAMN10061449 and SRR7866771. Raw reads were trimmed with TrimmomaticPE (*Bolger et al., 2014*) to remove low quality base calls. Trimmed reads were mapped to the *S. rosetta*

reference genome (*Fairclough et al., 2013*) using Burrows-Wheeler Aligner (*Li and Durbin, 2009*), and we removed PCR duplicates with Picard (http://broadinstitute.github.io/picard/). We realigned reads surrounding indel calls using GATK (*DePristo et al., 2011*) and called variants using SAMtools and bcftools (*Li et al., 2009*).

## Bulk segregant sequencing analysis

No large region of the genome (i.e. haplotype block) was found to co-segregate with the mutant phenotype in any of the crosses, likely because of the sparse, uneven distribution of genetic markers and/or high recombination rates. Sequence variants from the pooled samples were culled using vcftools vcf-isec (*Danecek et al., 2011*): (1) to keep only any sequence variants in the pooled samples that were shared with the parental mutant strain since any causative mutations should be present in both the pooled sample and the parental mutant strain, and (2) to remove any sequence variants in the pooled samples that were shared with the Mapping Strain (Isolate B), wild type (previously Isolate C), or the unmutagenized control from the Rosetteless mutagenesis (C2E5) since any of these sequence variants should not be causative for rosette defects (*Levin et al., 2014*; *Levin and King, 2013*). The remaining variants were filtered by quality: depth >2, quality score >10, and reference allele not N. The remaining list represents high quality variants in the pooled population that are shared with the mutant to the exclusion three different strains competent to form rosettes. Segregating variants were determined by dividing the number of reads that map to the alternative allele by the total number of high quality reads determined by SAMtools and bcftools (*Li et al., 2009*); any variants with >99% of reads that map to the alternative allele were considered variants that segregated with the mutant phenotype.

## Backcrosses

To test the linkage of clumpy phenotype and the predicted causative mutation from the bulk segregant analysis, F1s with the clumpy phenotype from the Jumble × Mapping Strain and Couscous ×Mapping Strain were backcrossed to the Mapping Strain. For the Jumble F1 backcross, $1 \times 10^6$ cells grown up from a clonally isolated F1 with the clumpy phenotype from Jumble ×Mapping Strain and $1 \times 10^6$ Mapping Strain cells were mixed, pelleted, and resuspended in 10 ml of 5% (v/v) *V. fischeri* conditioned media in ASW. After 24 hr, the *V. fischeri* conditioned media was replaced with 25% (v/v) CG media in ASW and cells were plated to limiting dilution. Clonally isolated thecate populations were genotyped by PCR of the microsatellite on supercontig 1 as described above and four heterozygous diploids populations were identified (probability of clonal isolation 0.79–0.95). The heterozygotes were rapidly passaged for 2 weeks to induce meiosis before being plated for clonal isolation (probability of clonal isolation 0.95–0.98). 12 F2s with the clumpy phenotype and 9 F2s with the rosette phenotype were identified (*Figure 2B*). Their DNA was extracted using the Base-Tris method described above and the region around the causal mutation was amplified. The resultant PCR product was digested for 4 hr with BfaI, which cleaves the mutant allele but not the wild type allele, and products of the digest were distinguished by agarose gel electrophoresis.

For the two Couscous F1 backcrosses, $2.5 \times 10^5$ cells from either one of two F1s with the clumpy phenotype from Couscous × Mapping Strain cross and $2.5 \times 10^5$ Mapping Strain cells were mixed, pelleted, resuspending in 0.5 ml of 2.5% (v/v) *V. fischeri* conditioned media in ASW. After 24 hr, *V. fischeri* conditioned media was replaced with 25% (v/v) CG media in ASW and cells were plated to limiting dilution (probability of clonal isolation 0.85–0.97). Clonally isolated thecate populations were genotyped by PCR of the microsatellite on supercontig 1 as described above and three heterozygous diploids (six total) were identified in each cross. Isolates were rapidly passaged for 2 weeks to induce meiosis before being plated for clonal isolation (probability of clonal isolation 0.88–0.97). 51 F2s with the clumpy phenotype and 38 F2s with the rosette phenotype were identified (*Figure 3B*); their DNA was extracted using the Base-Tris method described above, the region around the causal mutation was amplified, and the resultant PCR product was Sanger sequenced.

## Jumble and Couscous domain and structure prediction and alignment

Protein domains encoded by *jumble* (*Figure 2A*) and *couscous* (*Figure 3A*) were predicted using Interpro (*Finn et al., 2017*), PFAM (*Finn et al., 2016*), and the NCBI Conserved Domain Search

(*Marchler-Bauer et al., 2017*). Structural homology analysis of Jumble was performed with Phyre2 (*Kelley et al., 2015*) and HHphred (*Zimmermann et al., 2018*). The structure of the human N-acetyl-galactosaminyltransferase 4 (GlcNAc T4) catalytic domain (HHphred: E-value $7.5^{-19}$) was aligned to the predicted Jumble structure generated by HHphred using the PyMOL Molecular Graphics System, Version 2.0 Schrödinger, LLC (*Figure 2—figure supplement 2B*). Other choanoflagellate homologs of *jumble* were determined by reciprocal BLAST of the 20 sequenced choanoflagellate transcriptomes (*Richter et al., 2018*) and alignment was performed with ClustalX (*Larkin et al., 2007*) (*Figure 2—figure supplement 2A*). Four fungal homologs [*Saitoella complicata* (NCBI accession XP_019021578.1), *Dactylellina haptotyla* (NCBI accession EPS43829.1), *Naematelia encephala* (NCBI accession ORY22834.1), and *Tuber magnatum* (NCBI accession PWW71609.1)] were identified by best reciprocal BLAST using the *S. rosetta* Jumble protein sequence and aligned with ClustalX (*Larkin et al., 2007*) (*Figure 2—figure supplement 3*).The alignment of Couscous to yeast MNN2 glycosyltransferase domains was performed with ClustalX (*Larkin et al., 2007*) (*Figure 3—figure supplement 1A*).

## Generating transgenic constructs

Jumble (GenBank accession EGD72416/NCBI accession XM_004998928) and Couscous (GenBank accession EGD77026/NCBI accession XM_004990809) were cloned from wild type cDNA prepared as described in *Booth et al. (2018)*. Jumble^lw1 was cloned from cDNA prepared from the Jumble mutant. Couscous^lw1 could not be cloned from cDNA directly (possibly because of low mRNA levels due to nonsense mediate decay or simply because of high GC content of the gene). However, the 1 bp deletion in *Couscous^lw1* was confirmed by Sanger sequencing of genomic Couscous DNA. Site directed mutagenesis of the wild type gene was used to generate the mutant allele.

For complementation (*Figures 2C,D and* and *3C,D*), constructs were generated from a plasmid with a pUC19 backbone with a 5′ *S. rosetta* elongation factor L (*efl*) promoter, monomeric teal fluorescent protein (*mTFP*), and the 3′ UTR from actin (Addgene ID NK633) (*Booth et al., 2018*). A puromycin resistance gene was synthesized as a gene block and codon optimized for *S. rosetta*. The puromycin resistance gene (*pac*) was inserted after the *efl* promoter and separated from the fluorescent reporter by self-cleaving 2A peptide from the porcine virus (P2A) (*Kim et al., 2011*). Copies of *jumble*, *jumble^lw1*, *couscous,* and *couscous^lw1* were inserted either 5′ or 3′ of the mTFP and separated from mTFP by a flexible linker sequence (SGGSGGS) through Gibson cloning.

For fluorescent localization (*Figure 2E–H*, *Figure 2—figure supplement 4B*, *Figure 3—figure supplement 1B,C*), constructs were generated from a pUC19 backbone with a 5′ *S. rosetta* elongation factor L (*efl*) promoter, mWasabi, and 3′ UTR from actin. Copies of *jumble* (Addgene ID NK690), *jumble^lw1* (Addgene ID NK691), and *couscous* (Addegene ID NK692) were inserted either 5′ of the mWasabi separated by a flexible linker sequence (SGGSGGS) through Gibson cloning. Plasma membrane and ER markers from *Booth et al. (2018)* were used as previously described (Addgene ID NK624 and NK644).

## *S. rosetta* transfection and transgene expression

Transfection protocol was followed as described in *Booth et al. (2018)* (http://www.protocols.io/groups/king-lab). Two days prior to transfection, a culture flask (Corning, Cat. No. 353144) was seeded with Jumble, Couscous, or wild type cells at a density of 5,000 cells/ml in 200 ml of 5% (v/v) HN in AK. After 36–48 hr of growth, bacteria were washed away from the cells in three consecutive rounds of centrifugation and resuspension in sterile AK. After the final wash, the cells were resuspended in a total volume of 100 µl AK and counted on a Luna-FL automated cell counter (Logos Biosystems). The remaining cells were diluted to a final concentration of $5 \times 10^7$ cells/ml and divided into 100 µl aliquots. Each aliquot of cells pelleted at 2750 x g, resuspend in priming buffer (40 mM HEPES-KOH, pH 7.5; 34 mM Lithium Citrate; 50 mM L-Cysteine; 15% (w/v) PEG 8000; and 1 µM papain), and incubated at room temperature for 30 min to remove extracellular material coating the cells. Priming buffer was quenched with 50 mg/ml bovine serum albumin-fraction V (Sigma). Cells were pelleted at 1250 x g and resuspend in 25 µl of SF buffer (Lonza). Each transfection reaction was prepared by adding 2 µl of 'primed' cells to a mixture of 16 µl of SF buffer, 2 µl of 20 µg/ µl pUC19; 1 µl of 250 mM ATP, pH 7.5; 1 µl of 100 mg/ml Sodium Heparin; and 1 µl of each reporter DNA construct at 5 µg/µl. Transfections were carried out in 96-well nucleofection plate (Lonza) in a

Nucleofector 4d 96-well Nucleofection unit (Lonza) with the CM-156 pulse. Immediately after nucleo-fection, 100 μl of ice-cold recovery buffer (10 mM HEPES-KOH, pH 7.5; 0.9 M Sorbitol; 8% (w/v) PEG 8000) was added to the cells and incubated for 5 min. The whole volume of the transfection reaction plus the recovery buffer was transferred to 1 ml of 5% (v/v) HN in AK in a 12-well plate. After cells recovered for 1 hr, 5 μl of a 10 mg frozen *E. pacifica* pellet resuspend in 1 ml of AK was added to each well and RIFs were added if looking at rosette induction.

### Transgenic complementation

For complementation, Jumble mutants were transfected with the following plasmids: (1) *pEFl5'-Actin3'::pac-P2A-Jumble-mTFP* (Addgene ID NK694), (2) *pEFl5'-Actin3'::pac-P2A-Jumble$^{lw1}$-mTFP* (Addgene ID NK695), (3) *pEFl5'-Actin3'::pac-P2A-mTFP-Jumble* (Addgene ID NK696), (4) *pEFl5'-Actin3'::pac-P2A-mTFP-Jumble$^{lw1}$* (Addgene ID NK697), and (5) *pEFl5'-Actin3'::pac-P2A-mTFP* (Addgene ID NK676); and Couscous with the following plasmids: (1) *pEFl5'-Actin3'::pac-P2A-Cous-cous-mTFP* (Addgene ID NK698), (2) *pEFl5'-Actin3'::pac-P2A-Couscous$^{lw1}$-mTFP* (Addgene ID NK699), (3) *pEFl5'-Actin3'::pac-P2A-mTFP-Couscous* (Addgene ID NK700), (4) *pEFl5'-Actin3'::pac-P2A-mTFP-Couscous$^{lw1}$* (Addgene ID NK701), and (5) *pEFl5'-Actin3'::pac-P2A-mTFP* (Addgene ID NK676). Transformed cells were grown an additional 24 hr after transfection to allow for transgene expression, and then 40 μg/ml puromycin was added for selection. Selection occurred for 48 hr before rosette induction was counted by hemocytometer. After vortexing for 15 s and fixing with formaldehyde, 200 cells of each transfection well were counted on a hemocytometer to determine percentage of cells in rosettes (*Figure 2C*, *Figure 3C*). Complementation was repeated on two bio-logical replicates with three technical transfection replicates each. Representative rosette images (*Figure 2D*, *Figure 3D*) were taken on by confocal microscopy using Zeiss Axio Observer LSM 880 a C-Apochromat 40x/NA1.20 W Korr UV-Vis-IR water immersion objective.

### Live cell imaging

Glass-bottom dishes for live cell imaging were prepared by corona-treating and poly-D-lysine coat-ing as described in *Booth et al. (2018)*. Transfected cells were prepared for microscopy by pelleting 1–2 ml of cells and resuspend in 200 μl of 4/5 ASW with 100 mM LiCl to slow flagellar beating. Cells were plated on glass-bottom dishes and covered by 200 μl of 20% (w/v) Ficoll 400 dissolved in 4/5 ASW with 100 mM LiCl. Confocal microscopy was performed on a Zeiss Axio Observer LSM 880 with an Airyscan detector and a 63x/NA1.40 Plan-Apochromatic oil immersion objective.

Confocal stacks were acquired in super-resolution mode using ILEX line scanning and two-fold averaging and the following settings: 35 nm x 35 nm pixel size, 100 nm z-step, 0.9–1.0 μsec/pixel dwell time, 850 gain, 458 nm laser operating at 1–6% laser power, 561 nm laser operating at 1–2% laser power, 458/561 nm multiple beam splitter, and 495–550 nm band-pass/570 nm long-pass filter. Images were processed using the automated Airyscan algorithm (Zeiss).

### Lectin staining and jacalin quantification

All FITC labeled lectins from kits I, II, and III from Vector Lab (FLK-2100, FLK-3100, and FLK-4100) were tested for recognition in wild type, Jumbled, and Couscous (*Supplementary file - Table S5*). Cells were plated on poly-D-Lysine coated wells of a 96-well glass bottom plate, lectins were added at a concentration of 1:200 and imaged immediately using Zeiss Axio Observer.Z1/7 Widefield microscope with a Hammatsu Orca-Flash 4.0 LT CMOS Digital Camera and a 20x objective. For fur-ther jacalin image analysis (*Figure 4*), cells were plated on a poly-D-Lysine coated glass bottom dish, 1:400 FITC labelled jacalin and 1:200 lysotracker Red DN-99 (overloaded to visualize the cell body) and were imaged immediately by confocal microscopy using Zeiss Axio Observer LSM 880 a 63x/NA1.40 Plan-Apochromatic oil immersion objective. Images were taken with the following settings: 66 nm x 66 nm pixel size, 64 nm z-step, 0.34 μsec/pixel dwell time, 488 nm laser operating at 0.2% laser power with 700 master gain, and 561 nm laser operating at 0.0175% laser power with 750 mas-ter gain. Fifteen unique fields of view chosen based on lysotracker staining. Induced cells were treated with RIFs for 24 hr before imaging.

To process images, Z-stack images were max projected using ImageJ. Individual cells were cho-sen based on the ability to clearly see a horizontally oriented collar by lysotracker and cropped to only include a single cell. The maximum fluorescence intensity pixel of the jacalin channel was

determined for the cropped image and was used to normalize the fluorescence intensity. To measure jacalin staining around the cell body, a line was drawn using only the lysotracker staining from the point where the collar and the cell body meet on one side of the cell around the cell to the other and the fluorescence intensity was measured along the line. To compare between cells, the lines drawn around the cell body were one-dimensional interpolated in R to include 150 points and normalized to the length of the line. The average fluorescence intensity was plotted over the length of the line drawn around the cell body for Jumble, Couscous, and wild type -RIFs and +RIFs with a 95% confidence interval (*Figure 4F*). Measurements were taken from two biological replicates with at least 59 cells in total from each condition.

To examine jacalin localization for the Jumble and Couscous rescue experiments (*Figure 4—figure supplement 2*), FITC-conjugated jacalin could not be used due to its overlapping emission spectrum with the mTFP fusion protein used for complementation. Therefore, cells were incubated with 1 mg/ml biotinylated jacalin (Vector Labs, Cat. No. B-1155) for 5 min at room temperature and pelleted at 3000xg for 5 min. Once the supernatant was removed, the cells were incubated with 1:1000 Streptavidin Alexa Fluor 647 conjugate (Thermo Fisher Scientific, Cat. No. 32357) for 5 min at room temperature to fluorescently label the jacalin. The cells were then pelleted at 3000xg for 5 min, the supernatant was removed, and the cells were resuspended in ASW and plated for imaging. Jacalin localization was imaged by confocal microscopy using a Zeiss Axio Observer LSM 880 with a 63x/NA1.40 Plan-Apochromatic oil immersion objective.

## Wild type and mutant clumping assays

Wild type cells transfected with the puromycin resistance gene and mWasabi separated by the P2A self-cleaving peptide under the *efl* promoter and maintained in 40 µg/ml puromycin to enrich for positive transformants. For clumping assays, equal numbers of mWasabi-wt cells either without RIFs or treated with RIFs for 24 hr prior to the assay were mixed with either Jumble or Couscous, vortexed, and plated on BSA treated 8-well glass bottom dishes. DIC and fluorescent images were obtained after 30 min using Zeiss Axio Observer.Z1/7 Widefield microscope with a Hammatsu Orca-Flash 4.0 LT CMOS Digital Camera and a 40x/NA1.40 Plan-Apochromatic lens (*Figure 1—figure supplement 4*).

## Wild type and mutant growth curves

All cells strains were plated at a density of $1 \times 10^4$ cells/ml in 3 ml 5% (v/v) HN media in AK. Every 12 hr an aliquot of cells was vortexed vigorously for 15 s, fixed with formaldehyde, and counted by hemacytometer. Curves were generated from the average ±SD from two biological replicates with three technical replicates each (*Figure 1—figure supplement 2*).

## Jacalin western blot

Whole cell lysates were made from pelleting $1 \times 10^7$ cells at 4C at 3000 x g and resuspending in lysis buffer (20 mM Tris-HCl, pH 8.0; 150 mM KCl; 5 mM MgCl2; 250 mM Sucrose; 1 mM DTT; 10 mM Digitonin; 1 mg/ml Sodium Heparin; 1 mM Pefabloc SC; 0.5 U/µl DNaseI; 1 U/µl SUPERaseIN). Cells were incubated in lysis buffer for 10 min on ice and passed through a 30G needle 5x. Insoluble material was pelleted at 6000 x g for 10 min at 4C. Lysate ($1 \times 10^6$ cells/sample) was run on a 4–20% TGX mini-gel (Bio-Rad) for 45 min at 200 V and transferred onto 0.2 µm nitrocellulose membrane using the Trans-Blot Turbo Transfer System (Bio-Rad) with semi-dry settings 25V for min. The blot was blocked for 30 min with Odyssey PBS Block (Li-cor). The blot was probed with biotinylated jacalin (1:4,000; Vector Labs) and E7 anti-tubulin antibody (1:10,000; Developmental Studies Hybridoma Bank) diluted in block for 1 hr, and then with IRDye 800 streptavidin (1:1,000; Li-cor) and IRDye 700 mouse (1:1,000; Li-cor) in PBST [PBS with %1 Tween 20 (v/v)]. Blot was imaged on Licor Odyssey (*Figure 4—figure supplement 1*).

## Rosetteless immunofluorescence staining and imaging

Immunofluorescence (*Figure 4—figure supplement 3*) was performed previously described in *Levin et al. (2014)* with the modifications for better cytoskeleton preservation described in *Booth et al. (2018)*. Two ml of dense wild type, Jumble, and Couscous cells, that were either uninduced or induced with RIFs for 24 hr, were allowed to settle on poly-L-lysine coated coverslips (BD

Biosciences) for 30 min. Cells were fixed in two steps: 6% acetone in cytoskeleton buffer (10 mM MES, pH 6.1; 138 KCl, 3 mM MgCl$_2$; 2 mM EGTA; 675 mM Sucrose) for 5 min and then 4% formaldehyde diluted in cytoskeleton buffer for 20 min. The coverslips were gently washed three times with cytoskeleton buffer. Cells were permeabilized with permeabilization buffer [100 mM PIPES, pH 6.95; 2 mM EGTA; 1 mM MgCl$_2$; 1% (w/v) bovine serum albumin-fraction V; 0.3% (v/v Triton X-100)] for 30 min. Cells were stained with the anti-Rosetteless genomic antibody at 3.125 ng/μl (1:400), E7 anti-tubulin antibody (1:1000; Developmental Studies Hybridoma Bank), Alexa fluor 488 anti-mouse and Alexa fluor 647 anti-rabbit secondary antibodies (1:1000 each; Molecular Probes), and 6 U/ml rhodamine phalloidin (Molecular Probes) before mounting in Prolong Gold antifade reagent with DAPI (Molecular Probes).

Images were acquired on a Zeiss LSM 880 Airyscan confocal microscope with a 63x objective (as described for live cell imaging) by frame scanning in the super-resolution mode with the following settings: 30 nm x 30 nm pixel size; 100 nm z-step; 561 nm laser operating at 1.5% power with 700 master gain, and 488 nm laser operating at 2.0% power with 800 master gain. Wild type rosettes were imaged with 633 nm laser operating at 0.3% laser power and 650 master gain to prevent overexposure of Rosetteless, but all other conditions were operating at 2% laser power and 650 master gain in the 633 nm channel.

## Acknowledgements

Hannah Elzinga, Lily Helfrich, and Max Coyle helped with experiments and reagent preparation. We thank Iswar Hariharan and members of the King lab for helpful discussions, research support, mutant naming suggestions, and comments on the manuscript, especially Kayley Hake, Ben Larson, Tess Linden, and Thibaut Brunet. This work used the Vincent J Coates Genomics Sequencing Laboratory at UC Berkeley, supported by NIH S10 OD018174 Instrumentation Grant.

## Additional information

### Funding

| Funder | Grant reference number | Author |
| --- | --- | --- |
| Howard Hughes Medical Institute | | Laura A Wetzel<br>Tera C Levin<br>Ryan E Hulett<br>Daniel Chan<br>Grant A King<br>Reef Aldayafleh<br>David S Booth<br>Monika Abedin Sigg<br>Nicole King |
| Jane Coffin Childs Memorial Fund for Medical Research | Simons Fellow | David S Booth |

The funders had no role in study design, data collection and interpretation, or the decision to submit the work for publication.

### Author contributions

Laura A Wetzel, Conceptualization, Data curation, Formal analysis, Validation, Investigation, Visualization, Methodology, Writing—original draft, Writing—review and editing; Tera C Levin, Conceptualization, Investigation, Writing—review and editing; Ryan E Hulett, Daniel Chan, Reef Aldayafleh, Investigation; Grant A King, Investigation, Visualization; David S Booth, Conceptualization, Investigation, Methodology, Writing—review and editing; Monika Abedin Sigg, Conceptualization, Investigation, Methodology, Project administration, Writing—review and editing; Nicole King, Conceptualization, Supervision, Funding acquisition, Writing—original draft, Project administration, Writing—review and editing

## Author ORCIDs
Laura A Wetzel (iD) http://orcid.org/0000-0003-0391-2542
Tera C Levin (iD) https://orcid.org/0000-0001-7883-8522
Nicole King (iD) http://orcid.org/0000-0002-6409-1111

## Decision letter and Author response
Decision letter https://doi.org/10.7554/eLife.41482.054
Author response https://doi.org/10.7554/eLife.41482.055

# Additional files

## Supplementary files
• Supplementary file 1. Supplementary tables. (1) Table S1. Phenotypic classes of mutants isolated in this study and in the *Levin et al. (2014)* screen. (2) Table S2. Segregating variants in Rosetteless mapping cross. (3) Table S3. Segregating variants in Jumble mapping cross. (4) Table S4. Segregating variants in Couscous mapping cross. (5) Table S5. Fluorescent lectins tested.
DOI: https://doi.org/10.7554/eLife.41482.020

• Transparent reporting form
DOI: https://doi.org/10.7554/eLife.41482.021

## Data availability
Data have been deposited to the NCBI Sequence Read Archive under the project number PRJNA490902.

The following datasets were generated:

| Author(s) | Year | Dataset title | Dataset URL | Database and Identifier |
|---|---|---|---|---|
| Wetzel LA, Levin TC, Hulett RE, Chan D, King GA, Aldayafleh R, Booth DS, Sigg MA, King N | 2018 | Salpingoeca rosetta mutant and bulked segregant genome re-sequencing | https://www.ncbi.nlm.nih.gov/bioproject/?term=PRJNA490902 | NCBI BioProject, PRJNA490902 |
| Wetzel LA, Levin TC, Hulett RE, Chan D, King GA, Aldayafleh R, Booth DS, Sigg MA, King N | 2018 | Jumble mutant of Salpingoeca rosetta | https://www.ncbi.nlm.nih.gov/biosample/?term=SAMN10061445 | NCBI BioSample, SAMN10061445 |
| Wetzel LA, Levin TC, Hulett RE, Chan D, King GA, Aldayafleh R, Booth DS, Sigg MA, King N | 2018 | Couscous mutant of Salpingoeca rosetta | https://www.ncbi.nlm.nih.gov/biosample/?term=SAMN10061446 | NCBI BioSample, SAMN10061446 |
| Wetzel LA, Levin TC, Hulett RE, Chan D, King GA, Aldayafleh R, Booth DS, Sigg MA, King N | 2018 | Pooled F1s with mutant phenotype from Rosetteless x Mapping Strain Cross in Salpingoeca Rosetta | https://www.ncbi.nlm.nih.gov/biosample/?term=SAMN10061447 | NCBI BioSample, SAMN10061447 |
| Wetzel LA, Levin TC, Hulett RE, Chan D, King GA, Aldayafleh R, Booth DS, Sigg MA, King N | 2018 | Pooled F1s with mutant phenotype from Jumble x Mapping Strain Cross in Salpingoeca Rosetta | https://www.ncbi.nlm.nih.gov/biosample/?term=SAMN10061448 | NCBI BioSample, SAMN10061448 |
| Wetzel LA, Levin TC, Hulett RE, Chan D, King GA, Aldayafleh R, Booth DS, Sigg MA, King | 2018 | Pooled F1s with mutant phenotype from Couscous x Mapping Strain Cross in Salpingoeca Rosetta | https://www.ncbi.nlm.nih.gov/biosample/?term=SAMN10061449 | NCBI BioSample, SAMN10061449 |

N

| | | | | |
|---|---|---|---|---|
| Wetzel LA, Levin TC, Hulett RE, Chan D, King GA, Aldayafleh R, Booth DS, Sigg MA, King N | 2018 | Seafoam mutant gDNA sequencing | https://www.ncbi.nlm.nih.gov/biosample/?term=SAMN10501893 | NCBI BioSample, SAMN10501893 |
| Wetzel LA, Levin TC, Hulett RE, Chan D, King GA, Aldayafleh R, Booth DS, Sigg MA, King N | 2018 | Soapsuds mutant gDNA sequencing | https://www.ncbi.nlm.nih.gov/biosample/?term=SAMN10501894 | NCBI BioSample, SAMN10501894 |
| Wetzel LA, Levin TC, Hulett RE, Chan D, King GA, Aldayafleh R, Booth DS, Sigg MA, King N | 2018 | Jumble mutant gDNA sequencing | https://www.ncbi.nlm.nih.gov/sra/?term=SRR7866767 | NCBI Sequence Read Archive, SRR7866767 |
| Wetzel LA, Levin TC, Hulett RE, Chan D, King GA, Aldayafleh R, Booth DS, Sigg MA, King N | 2018 | Couscous mutant gDNA sequencing | https://www.ncbi.nlm.nih.gov/sra/?term=SRR7866768 | NCBI Sequence Read Archive, SRR7866768 |
| Wetzel LA, Levin TC, Hulett RE, Chan D, King GA, Aldayafleh R, Booth DS, Sigg MA, King N | 2018 | Rosetteless x Mapping Strain cross gDNA sequencing | https://www.ncbi.nlm.nih.gov/sra/?term=SRR7866769 | NCBI Sequence Read Archive, SRR7866769 |
| Wetzel LA, Levin TC, Hulett RE, Chan D, King GA, Aldayafleh R, Booth DS, Sigg MA, King N | 2018 | Jumble x Mapping Strain cross gDNA sequencing | https://www.ncbi.nlm.nih.gov/sra/?term=SRR7866770 | NCBI Sequence Read Archive, SRR7866770 |
| Wetzel LA, Levin TC, Hulett RE, Chan D, King GA, Aldayafleh R, Booth DS, Sigg MA, King N | 2018 | Couscous x Mapping Strain cross gDNA sequencing | https://www.ncbi.nlm.nih.gov/sra/?term=SRR7866771 | NCBI Sequence Read Archive, SRR7866771 |
| Wetzel LA, Levin TC, Hulett RE, Chan D, King GA, Aldayafleh R, Booth DS, Sigg MA, King N | 2018 | Soapsuds mutant gDNA sequencing | https://www.ncbi.nlm.nih.gov/sra/?term=SRR8263909 | NCBI Sequence Read Archive, SRR8263909 |
| Wetzel LA, Levin TC, Hulett RE, Chan D, King GA, Aldayafleh R, Booth DS, Sigg MA, King N | 2018 | Seafoam mutant gDNA sequencing | https://www.ncbi.nlm.nih.gov/sra/?term=SRR8263910 | NCBI Sequence Read Archive, SRR8263910 |

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
