## [Decision Letter]

Thank you for submitting your article "Glycosyltransferases promote development and prevent promiscuous cell aggregation in the choanoflagellate *S. rosetta*" for consideration by *eLife*. Your article has been reviewed by two peer reviewers, and the evaluation has been overseen by a Reviewing Editor and Marianne Bronner as the Senior Editor. The following individual involved in review of your submission has agreed to reveal his identity: Iñaki Ruiz-Trillo (Reviewer #1).

The reviewers have discussed the reviews with one another and the Reviewing Editor has drafted this decision to help you prepare a revised submission.

Summary:

This Research Advance manuscript builds on past findings published in *eLife* by the same group that identified and characterized the Rosetteless gene in the choanoflagellate *S. rosetta*. Here, the authors report on a genetic screen in the choanoflagellate *Salpingoeca rosetta* aimed at identifying genes essential for the development of choanoflagellate colonies. Using a bulk segregant assay, two genes (Jumble and Couscous) were identified encoding presumptive glycosyltransferases. The sequencing, as well as the mapping data and complementation studies are sound. The mutants disrupt the normal pattern of rosettes and instead result in clumps of cells. The authors conclude that their findings suggest a pre-metazoan role for glycosyltransferases in regulating development.

Essential revisions:

1) A major concern is regarding the aggregative behavior of Jumble and Couscous mutants. It is unclear whether the data presented demonstrate that Jumble and Couscous develop by aggregation. It is evident that once they are disaggregated, they aggregate to form the clumps again. But what about the original clumps? Are they really formed by aggregation? Maybe we missed something, but could it not be possible that the original Couscous and Jumble clumps developed by clonal division? This needs to be more clearly explained.

2) Another area that could be improved upon is the lectin localization data.

The mis-localization of the lectin jacalin in the Jumble and Couscous mutants is interesting. Is that localization restored to wild-type by complementation? Or linked to the mutations following a genetic cross? What about in rosetteless mutants or in the two other mutants that could not be mapped here, Soapsuds and Seafoam? Absent complementation data, or genetic linkage data, or congruent phenotype of two independent mutants, an observed mutant phenotype might or might not be linked to the suspect causative mutation.

---

## [Author Response]

Essential revisions:1) A major concern is regarding the aggregative behavior of Jumble and Couscous mutants. It is unclear whether the data presented demonstrate that Jumble and Couscous develop by aggregation. It is evident that once they are disaggregated, they aggregate to form the clumps again. But what about the original clumps? Are they really formed by aggregation? Maybe we missed something, but could it not be possible that the original Couscous and Jumble clumps developed by clonal division? This needs to be more clearly explained.

To investigate this further, we tested the necessity of cell division for clump formation in Jumble and Couscous mutant cells using the cell cycle inhibitor, aphidicolin. Aphidicolin prevented cell division but did not prevent clump formation in Jumble and Couscous (see new Figure 1—figure supplement 1). Thus, cell division is not required for clump formation. While it is formally possible, and indeed likely, that cell division contributes partially to clump *growth* over longer periods of time (>6hr, the time required for cell doubling) as cells within clumps divide, aggregation alone is sufficient for both the initiation and growth of clumps.

We added Figure 1—figure supplement 1 and revised the text to read:

“Moreover, blocking cell division with the cell cycle inhibitor aphidicolin did not prevent clump formation (Figure 1—figure supplement 1). […] Indeed, each of the mutants tested also displayed a mild defect in cell proliferation (Figure 1—figure supplement 2)."

2) Another area that could be improved upon is the lectin localization data.The mis-localization of the lectin jacalin in the Jumble and Couscous mutants is interesting. Is that localization restored to wild-type by complementation? Or linked to the mutations following a genetic cross? What about in rosetteless mutants or in the two other mutants that could not be mapped here, Soapsuds and Seafoam? Absent complementation data, or genetic linkage data, or congruent phenotype of two independent mutants, an observed mutant phenotype might or might not be linked to the suspect causative mutation.

To test whether the mutations in *jumble* and *couscous* were causative for the jacalin mislocalization observed in the Jumble and Couscous mutants (Figure 4), we examined the localization of jacalin in transgenically complemented rosettes.

Transformation of Jumble and Couscous with a negative control – *mTFP –* did not restore jacalin staining at the basal pole of Jumble and Couscous cells (see new Figure 4—figure supplement 2G, I), but rosettes recovered through complementation with either transgenic *mTFP-jumble* or *couscous-mTFP* had displayed strong jacalin staining in the center of rosettes (Figure 4—figure supplement 2H, J). Thus, *jumble* and *couscous* are sufficient to restore wild type glycosylation pattern in complemented rosettes.

We added Figure 4—figure supplement 2 and revised the text to read:

“Transformation of Jumble cells with *mTFP-jumble* not only rescued rosette development (Figure 2C, D), but also restored the wild type glycosylation pattern, as revealed by jacalin staining in the center of complemented rosettes (Figure 4—figure supplement 2). […] Thus, the glycosylation defects in Jumble and Couscous mutant cells were directly linked to the genetic lesions in *jumble* and *couscous,* respectively.”